# Mapping autophagosome contents identifies interleukin-7 receptor-α as a key cargo modulating CD4+ T cell proliferation

Dingxi Zhou[1], Mariana Borsa ®[1], Daniel J. Puleston[1], Susanne Zellner ®[2], Jesusa Capera ®[1], Sharon Sanderson[1], Martina Schifferer ®[3,4], Svenja S. Hester[5], Xin Ge ®[1], Roman Fischer ®[5,6], Luke Jostins ®[1], Christian Behrends ®[2,9], Ghada Alsaleh[1,7,9] & Anna Katharina Simon ®[1,8,9] ✉

CD4+ T cells are pivotal cells playing roles in the orchestration of humoral and cytotoxic immune responses. It is known that CD4+ T cell proliferation relies on autophagy, but identification of the autophagosomal cargo involved is missing. Here we create a transgenic mouse model, to enable direct mapping of the proteinaceous content of autophagosomes in primary cells by LC3 proximity labelling. Interleukin-7 receptor-α, a cytokine receptor mostly found in naïve and memory T cells, is reproducibly detected in autophagosomes of activated CD4+ T cells. Consistently, CD4+ T cells lacking autophagy show increased interleukin-7 receptor-α surface expression, while no defect in internalisation is observed. Mechanistically, excessive surface interleukin-7 receptor-α sequestrates the common gamma chain, impairing the interleukin-2 receptor assembly and downstream signalling crucial for T cell proliferation. This study shows that key autophagy substrates can be reliably identified in this mouse model and help mechanistically unravel autophagy's contribution to healthy physiology and disease.

Macroautophagy, hereafter as autophagy, is an evolutionarily conserved catabolic pathway that mediates the degradation of cellular components. Autophagy plays an essential role in the differentiation, homeostasis and renewal of many immune cells[1]. Among these immune populations is the CD4+ T cell, or helper T cell, which orchestrates both adaptive and innate immunity. To exert their helper function, CD4+ T cells need to be activated and differentiated into effector populations. The activity is finely tuned by three sequential signals: (1) T cell receptor (TCR) recognising their cognate antigen in a major histocompatibility complex (MHC)-restricted manner, (2) costimulatory receptors activated by their ligands, and (3) cytokine-amplified differentiation and expansion[2]. Moreover, activated T cells undergo a dramatic shift in their proteome, which facilitates their functional and metabolic transition[3]. Autophagy has been shown to contribute to the expansion, differentiation and maintenance of CD4+ T cells, suggesting that autophagy can selectively degrade a spectrum of molecules to mediate these physiological processes[4–6]. The systematic identification of autophagosomal cargoes in

[1]The Kennedy Institute of Rheumatology, Nuffield Department of Orthopaedics Rheumatology & Musculoskeletal Sciences, University of Oxford, Roosevelt Drive, OX3 7FY Oxford, UK. [2]Munich Cluster for Systems Neurology (SyNergy), Medical Faculty, Ludwig-Maximilians-University München, Feodor-Lynen Strasse 17, 81377 Munich, Germany. [3]German Center for Neurodegenerative Diseases (DZNE), 81377 Munich, Germany. [4]Munich Cluster for Systems Neurology (SyNergy), 81377 Munich, Germany. [5]Nuffield Department of Medicine, Target Discovery Institute, University of Oxford, OX3 7FZ Oxford, UK. [6]Chinese Academy of Medical Science (CAMS) Oxford Institute (COI), University of Oxford, OX3 7FZ Oxford, UK. [7]Botnar Research Centre, Nuffield Department of Orthopaedics Rheumatology & Musculoskeletal Sciences, University of Oxford, OX3 7LD Oxford, UK. [8]Max Delbrück Center for Molecular Medicine, Robert-Rössle-Straße 10, 13125 Berlin, Germany. [9]These authors contributed equally: Christian Behrends, Ghada Alsaleh, Anna Katharina Simon. ✉e-mail: katja.simon@imm.ox.ac.uk

CD4+ T cells will fill a gap in our understanding of the molecular mechanisms of T cell activation and proliferation.

Traditional methods for mapping autophagosomal content are based on purifying autophagosomes with extensive cell fractionation and mass-spectrometry analysis. However, the results are compromised by contamination from other co-purified cellular compartments and by the low reproducibility caused by variation introduced during the differential centrifugation and extensive manipulation. Furthermore, the number of cells required often exceeds what can be obtained from primary material. To determine bona fide autophagosomal constituents, the proximity-dependent biotinylation approach provides a solution. It uses a modified ascorbate peroxidase (APEX2) to transform biotin-phenols into short-lived reactive radicals, which can covalently tag proteins in close proximity[7]. When genetically fused to a protein of interest, the enzyme provides a snapshot of the local environment and maps the spatial proteome of subcellular compartments[8]. Le Guerroué et al.[9] employed the technique to profile potential autophagosomal substrate proteins in HeLa cells through overexpressing APEX2 fused with human ATG8.

In this work, to extend its application to primary cells under different physiological contexts, we create a knock-in mouse model by introducing the APEX2 sequence into the endogenous *Lc3b* locus, an *Atg8* homologue in mammalian cells. This model helps us to identify IL-7Rα as a major autophagy cargo in proliferating CD4+ T cells, thus revealing how autophagy contributes to CD4+ T cell expansion by modulating signal 3.

## Results

### Autophagy-deficient CD4+ T cells show delayed proliferation
In a study by Ahmed et al. and our previous study, it was revealed that autophagy is crucial for the formation of memory CD8+ T cells[10,11]. We used a mouse model conditionally deleting *Atg7*, a key autophagy gene, in both CD4+ and CD8+ T lymphocytes (CD4[cre], *Atg7[flox/flox]*, hereafter as T-Atg7[−/−]). Using the same mouse model here, we addressed the role of autophagy in CD4+ T cells and found that the formation of CD4+ effector T cells was severely affected in two models of antigen exposure (Fig. 1a–d). We excluded the effects of autophagy deletion in antigen-presenting cells expressing CD4 by generating chimaera with bone marrow from wild-type littermate controls (CD45.1+) and T-Atg7[−/−] mice (CD45.2+) (Fig. 1a). Hosts were challenged with murine cytomegalovirus (MCMV) after 7 weeks and MCMV-specific CD4+ T cells in both spleens and lungs were evaluated with MHC class II tetramers, on day 7–9, the peak time for effector T cell formation[12]. The number of CD45.2+ MCMV-specific T-Atg7[−/−] cells was significantly lower than that of CD45.1+ wild-type cells (spleens in Fig. 1b and lungs in Supplementary Fig. 1a).

To further confirm that autophagy-deficient CD4+ T cells fail to expand upon antigenic stimulation, we bred the T-Atg7[−/−] mice with the OT-II T-cell receptor transgenic mouse model, where T cells specifically recognise an epitope spanning peptide residues 329–337 of ovalbumin (OVA)[13]. Neither CD4+ T cell lymphopenia nor increase of effector memory population (CD62L⁻ CD44[high]) were observed in the spleens of naïve 6-week-old T-Atg7[−/−] OT-II mice (Supplementary Fig. 1b–c). We transferred either wild-type or T-Atg7[−/−] OT-II T cells of CD45.2+ background into CD45.1+ host mice. After 24 h, mice were challenged with recombinant H5 adenovirus expressing OVA (AdH5-OVA) to activate the OT-II T cells (Fig. 1c). On day 9, phenotyping of antigen-specific cells revealed that autophagy-deficiency significantly compromised effector T cell formation in both spleen and blood (Fig. 1d and Supplementary Fig. 1d), which is consistent with our findings measuring endogenous T cell responses to MCMV.

To understand why antigen-specific CD4+ T cells show impaired expansion in vivo, we next dissected the dynamics of the TCR-mediated proliferation in autophagy-deficient cells. We activated wild-type and autophagy-deficient OT-II CD4+ T cells in vitro and assessed their proliferation profile. We observed a delay in proliferation of T-Atg7[−/−] CD4+ T cells compared with their wild-type counterparts on day 3 and day 7 (Fig. 1e and Supplementary Fig. 1e). However, by day 10, autophagy-deficient CD4+ T cells caught up and exhibited a similar profile to wild-type. In line with previous work, the upregulation of early activation markers is not significantly changed between autophagy-deficient and -proficient cells, indicating that the TCR-signalling is not affected (Supplementary Fig. 1f, g)[14].

To rule out the preferential proliferation (and escape) of CD4+ T cells with incomplete *Atg7* knockout, we tested *Atg7* mRNA expression by qRT-PCR on day 7 in sorted T cells that had undergone a maximum number of detectable cell divisions (i.e. diluted the cell dye to the maximum). Indeed, *Atg7* mRNA in proliferated CD4+ T cells from T-Atg7[−/−]; OT-II mice was undetectable (Supplementary Fig. 1h). Moreover, decreased expansion cannot be solely explained by increased cell death, since the percentages of CD4+ T cells did not differ significantly between wild-type and knockout cells on day 1 post activation, and it only displayed a significant but mild increase on day 3 (Supplementary Fig. 1i). Since the survival of activated T cells is modulated by multiple signalling pathways[15,16], this also implies certain deficits in these pathways when cells are lacking autophagy. Our observation of delayed proliferation in response to antigen-stimulation confirms and extends the findings of other autophagy-knockout models that demonstrated an issue with CD4+ T cell expansion in response to non-specific stimulation[14,17–20]. Next, we aimed to identify the proteins that are selectively degraded by autophagy by using proximity labelling.

### A mouse model based on the proximity biotinylation technique
Among the six mammalian ATG8 homologues, MAP1LC3B (LC3B) is the most widely used to detect autophagic flux. Furthermore, in the study by Le Guerroué et al., autophagosomal substrates labelled by LC3B-fused APEX2 displayed the highest reproducibility between biological samples among all six ATG8 homologues[9]. Therefore, we created a transgenic mouse model, knocking the sequence of APEX2 (AP2) and a flexible GS-linker into the endogenous *Lc3b* locus (Fig. 2a). In the presence of both biotin-phenol (BP) and hydroxyl-peroxide (H₂O₂), LC3B-fused AP2 transforms biotin-phenol into highly reactive radicals, which can further be covalently conjugated to proteins nearby (Fig. 2a). Then, these biotinylated proteins undergo affinity purification and mass spectrometry-based profiling. LC3B is expressed on the cytosolic and luminal side of autophagosomes but only on the cytosolic side of LC3-associated phagosomes[21,22]. To increase the chance of discovering targets on the luminal side of autophagosomes, homogenates extracted from the proximity-labelled cells, which contains intact autophagosomes and other organelles, were incubated with proteinase K (Prot K) (Fig. 2a). This allows Prot K to degrade the cytosolic proteins bound to LC3B, while those residing inside autophagosomes and other vesicles remained largely intact due to the protective effect of the membranes[9].

Wild-type, heterozygous and homozygous mice are identified by genomic PCR (Supplementary Fig. 2a). We first confirmed that the LC3B-AP2 chimaeric protein is expressed. Cellular LC3B has two major isoforms – the free LC3B-I located in the cytosol and the membrane-bound LC3B-II. Immunoblotting of splenocytes showed bands around 44 kDa (LC3B: 17 kDa, AP2: 27 kDa) in heterozygotes and homozygotes with anti-LC3B staining, but not in wild-type cells (Fig. 2b). In wild-type and heterozygous splenocytes, LC3B was mostly expressed as non-fused to AP2. As expected, the lipid-conjugated form of LC3B-AP2 (LC3B-II-AP2) and non-fused LC3B-II accumulated when cells were treated with bafilomycin A1 (BafA1), an inhibitor preventing the lysosomal degradation of LC3 in autophagosomes. These results confirm that LC3B-fused AP2 is targeted to autophagosomes and undergoes lysosomal degradation, as expected from a functional LC3B protein. Next, we determined the enzymatic activity of AP2. After 30 min BP

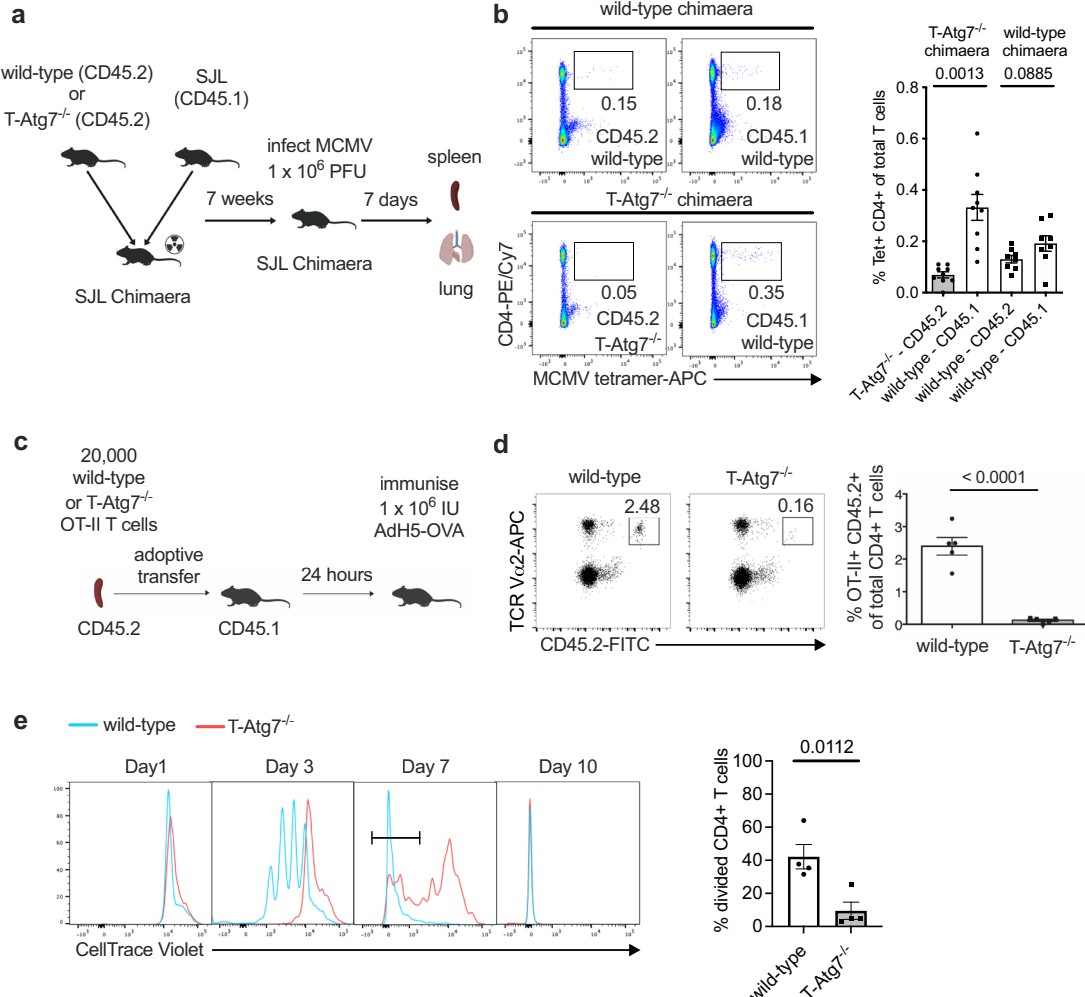

**Fig. 1 | Autophagy-deficient CD4+ T cells show delayed proliferation.**
**a** Experimental set-up for the generation of bone marrow (BM) chimaeras. Lethally irradiated CD45.1+ hosts were reconstituted with a 1:1 mix of BM of either T-Atg7$^{-/-}$ or wild-type (donors both CD45.2+) and CD45.1+ wild-type BM. After 7 weeks, hosts were intravenously (i.v.) infected with $1 \times 10^6$ PFU murine cytomegalovirus (MCMV). Cells from spleens were collected and analysed 7 days post-infection. **b** Dot plots (left) of splenic MCMV-specific CD4+ T cells within CD19- population from CD45.2+ wild-type or T-Atg7$^{-/-}$ donors and CD45.1+ donors. Bar graph (right) indicates percentage of tetramer (Tet)+ CD4+ T cells within the CD45.2+ CD19- or CD45.1+ CD19- population ($n = 8$ mice for wild-type chimaera group, $n = 9$ mice for T-Atg7$^{-/-}$ chimaera group), represented as mean ± SEM with paired two-tailed Student's $t$ test. **c** Experimental set-up for the adoptive transfer: OT-II lymphocytes ($2 \times 10^4$) from either CD45.2 wild-type or T-Atg7$^{-/-}$ mice were adoptively transferred into CD45.1 recipient mice. The recipient mice were orally immunised with $1 \times 10^6$ IU

recombinant H5 adenovirus expressing ovalbumin (AdH5-OVA). The number of OT-II cells was determined by flow cytometry 8 days after immunisation. **d** Dot plots (left) are gated on CD45.2+, TCR Vα2+ in the spleens of recipient mice. Bar graph (right) depicts the frequency of gated population within total CD4+ T cell population ($n = 5$ mice per group). **e** Splenic OT-II+ T cells from wild-type or T-Atg7$^{-/-}$ mice were stimulated with ovalbumin in culture medium supplemented with murine IL-2 for 1, 3, 7, or 10 days. Histograms by flow cytometry (left) represent OT-II + CD4+ T cell proliferation. Bar graph (right) indicates the percentage of Cell-Trace Violet-negative cells (most divided, defined by gate on day 7) within total OT-II+CD4+ T cells ($n = 4$ mice per group). All data are representative of three independent experiments. **d, e** are represented as mean ± SEM with unpaired two-tailed Student's $t$ test. Exact $p$ values are depicted in the figure. Source data are provided as a Source Data file.

pre-incubation and 1 min $H_2O_2$ treatment, mouse splenocytes were lysed and analysed by western blotting with streptavidin. Cellular lysates from homozygote mice (two copies of *Lc3b-AP2*) treated with BP and $H_2O_2$ showed the strongest bands spanning from 11 kDa to 200 kDa, whereas negative controls in which we omitted BP or $H_2O_2$, or cell lysates from mice that do not express LC3B-AP2, showed only weak biotin bands (Supplementary Fig. 2b).

To optimise the protocol of autophagosomal protein purification, we immortalised mouse embryonic fibroblasts (imMEFs) extracted from *Lc3b-AP2* embryos by transfecting them with lentivirus expressing SV40 antigens. In the imMEFs, as expected, we observed the co-localisation of biotinylated proteins with AP2 and LAMP1, a lysosomal marker (Fig. 2c), further confirming the autophagosomal enrichment of LC3B-AP2. Biotin largely co-localises with AP2 (Pearson's coefficient

over 0.8; Supplementary Fig. 2c) which indicates proteins labelled with biotin are proximal to AP2 enzymes. In contrast, since not all autophagosomes are fused with lysosomes at once, biotin was only partially co-localised with LAMP1 (Supplementary Fig. 2c, Pearson's coefficient around 0.3). As expected, when imMEFs were treated with BafA1, the colocalisation between these two molecules was significantly decreased. In addition, the LC3B-AP2 chimaeric protein generated electron microscope-dense autophagosomal structures when cells were subjected to 3,3-diaminobenzidine labelling and $H_2O_2$ pulsing, which were more prominent with BafA1-treatment (Fig. 2d). To further validate that the LC3B-AP2 chimaeric protein is indeed located on the autophagosomal inner membrane, thus labelling luminal proteins, we performed Prot K protection assays. Homogenates from proximity-biotinylated imMEFs were incubated with Prot K and/or Triton-X, a

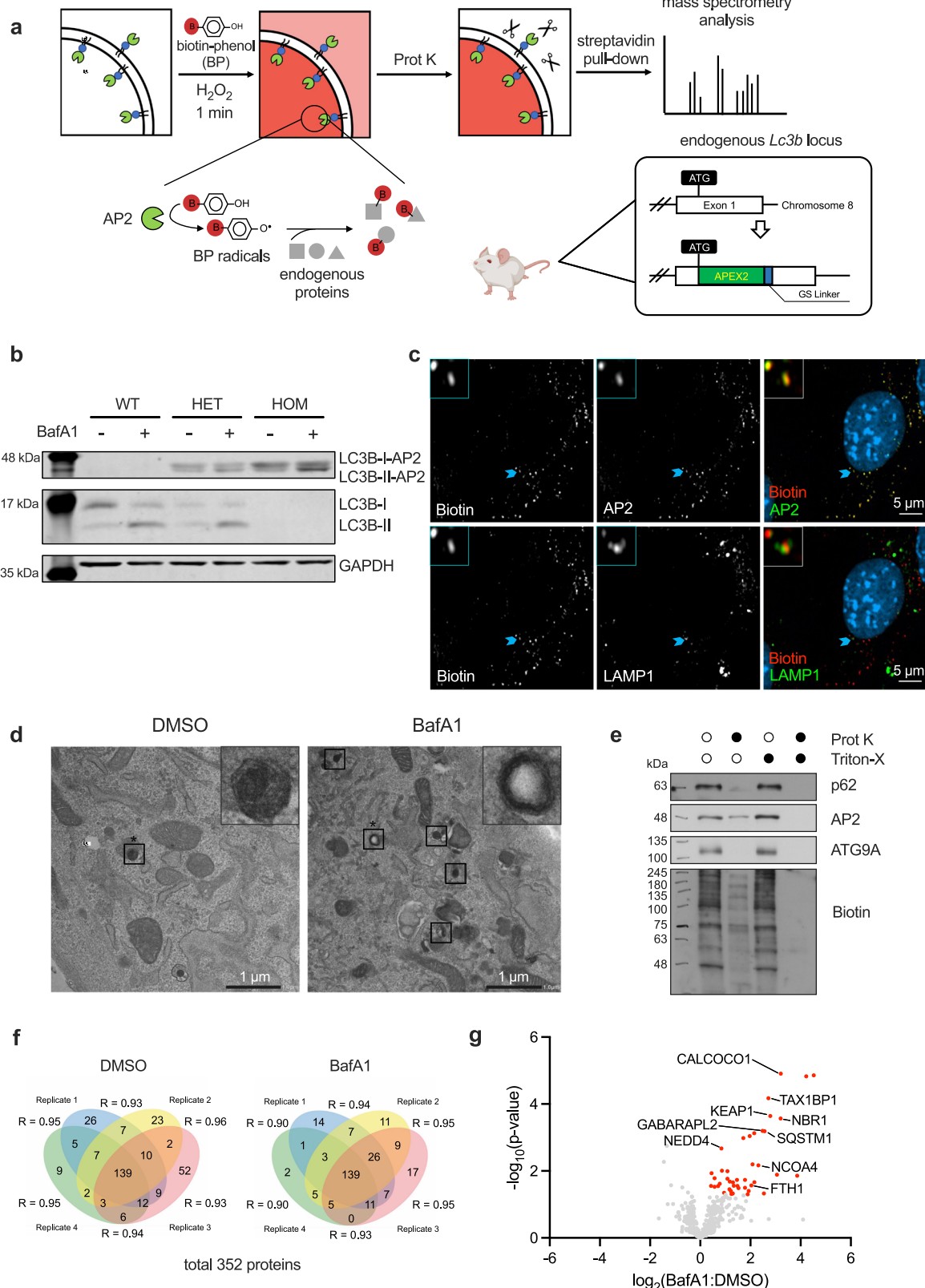

detergent that breaks all membrane structures. p62, an autophagy cargo receptor protein, and LC3-AP2 (and their proximity-biotinylated proteins) are located both inside and outside of autophagosomes. Prot K can only digest the proteins outside the autophagosomes, having no access to the inside ones which are protected by the intact double-membrane structure. Therefore, and similar to previous observations in LC3B-AP2 overexpression systems in vitro[9,23], the sample treated

with only Prot K partially preserved p62, LC3-AP2 and biotinylated bands compared to untreated or Triton-X treated control, while the cytosolic tail of the transmembrane protein ATG9A was completely degraded (Fig. 2e). By contrast, adding both Prot K and Triton-X removed all proteins. Together, these results indicate the correct localisation of LC3B-AP2 proteins in autophagosomes and the protective effect of the autophagosomal membranes to Prot K.

**Fig. 2 | LC3B-AP2 transgenic mouse model facilitates the direct identification of autophagosomal cargoes. a** Generation of mouse model and experimental set-up for proximity labelling. **b** Western blot of whole splenic cell homogenates from wild-type, heterozygous and homozygous mice. Cells were treated with 10 nM bafilomycin A1 (BafA1) or DMSO for 2 h. These experiments were repeated as biological triplicates (one mouse each) with similar results. **c** Confocal imaging of biotinylated proteins, AP2 and LAMP1 in proximity-labelled immortalised mouse embryonic fibroblasts (imMEFs) generated from the *Lc3b-AP2* mouse model. Light blue arrows indicate co-localisation events with magnifications at the upper-left corners. These experiments were repeated as biological triplicates with similar results. **d** Electron micrographs of DMSO- or BafA1-treated *Lc3b-AP2* imMEFs. Following fixation, cells were incubated with 3,3-diaminobenzidine and $H_2O_2$ prior to standard embedding and ultrathin sectioning. Squares highlight the autophagosomal structures. Squares that are marked by asterisks are magnified in the upper-right corner. **e** Homogenates from imMEFs were labelled, then left untreated or incubated with Prot K, Triton-X, or both and followed by immunoblotting for AP2, Biotin, p62 and ATG9A. **f** Venn diagrams depicting Prot K-protected, biotinylated proteins identified in imMEFs with more than two spectral counts across four biological replicates in both DMSO and BafA1-treated groups. Pearson's correlation coefficients of the label-free quantity of each protein included in two replicates are displayed. **g** Volcano plot of proteins from **f** labelled by LC3B-AP2 in imMEFs. Proteins significantly upregulated in response to BafA1 treatment are highlighted in red ($p < 0.05$ by unpaired two-tailed Student's *t* test, $n = 4$ biological replicates). Known autophagosomal candidates are labelled with protein names. Source data are provided as a Source Data file.

To test whether our mouse model can indeed help identify autophagy substrates, we next performed quantitative mass spectrometry-based proteomics using *Lc3b-AP2* imMEFs (Supplementary Fig. 2d). Cells were treated or not with BafA1 and four biological replicates were included in each group. After excluding Prot K-resistant candidates found when cells were treated with MS-compatible detergent RAPIGest, we reproducibly identified a total of 352 proteins specifically protected from Prot K treatment (Fig. 2f and Supplementary Data 1)[23]. Among the Prot K-protected proteins, 43 proteins were significantly increased upon BafA1 treatment (Fig. 2g). Gene ontology analysis revealed that the identified proteins belong to the category of proteins that are usually enriched in autophagosomes and other cytoplasmic vesicles, where the membrane-bound LC3B is mainly located (Supplementary Fig. 2e)[24]. As expected, most of the proteins identified under basal conditions include autophagic receptors (p62/SQSTM1, NBR1, NCOA4, CALCOCO1 and TAX1BP1), autophagy regulators interacting with LC3 (NEDD4), autophagy substrates (KEAP1, FTH1) and another ATG8 homologue (GABARAPL2). Taken together, these observations support the reliability of our mouse model, which can be used as a tool to specifically identify autophagosomal cargoes.

To determine the minimum cell number to detect autophagosomal cargo effectively, we performed a cell titration experiment with cell numbers ranging from 0.125 to 8 million. Although two autophagic receptors, TAX1BP1 and p62/SQSTM1, could be consistently identified as BafA1-sensitive candidates in all conditions, the total number of proteins being significantly upregulated by BafA1 showed a sharp decrease when less than 1 million imMEFs were used (Supplementary Fig. 2f). These data indicate that a greater cell number potentially yields more candidates when using the LC3B-AP2 labelling system.

### IL-7Rα is detected in LC3-containing vesicles in activated CD4+ T cells

Aiming to better understand the underlying causes of the observed delayed proliferation in autophagy-deficient CD4+ T cells, we decided to apply the LC3B-AP2 labelling system to investigate the autophagosomal degradome in autophagy-proficient proliferative CD4+ T cells. For that, we used activated CD4+ T cells isolated from *Lc3b-AP2* mice. The proliferative state of CD4+ T cells allowed us to obtain a large number of fully activated CD25+ cells, which, based on the results obtained from imMEFs, contributes to the identification of higher numbers of autophagosomal candidates (Supplementary Fig. 3a). In addition, we could determine that the autophagic flux was highest on days 2 and 3 (Supplementary Fig. 3b), indicating these are appropriate time points to evaluate the role of autophagy in CD4+ T cell proliferation. Finally, we identified an accumulation of LC3B-II-AP2 protein in BafA1-treated cells 3 days post-activation (Supplementary Fig. 3c). Thus, we decided to activate CD4+ T cells in vitro for 3 days, using anti-CD3/CD28 Dynabeads, prior to performing proximity labelling.

Again, the mass-spectrometry analysis showed high reproducibility between each biological replicate (Fig. 3a). Among 112 Prot K-protected and streptavidin-enriched proteins, seven candidates were significantly upregulated by BafA1-treatment (Fig. 3b and Supplementary Data 2). Of these candidates, interleukin-7 receptor-α (IL-7Rα) showed the highest fold change robustly among all biological replicates (4.67-fold increase, $p = 0.033$, Fig. 3c). Moreover, IL-7Rα was found to be significantly upregulated ($p = 0.0089$) in a repeat experiment, and a meta-analysis combining the two experiments found that IL-7Rα was the only protein that was significant after multiple testing correction (meta-analysed $p = 0.00087$, Benjamini-Hochberg-corrected $q$ value = 0.0315, Supplementary Fig. 3d and Supplementary Data 3). Therefore, we decided to further investigate this protein. IL-7Rα is highly expressed on naive T cells, and TCR/CD28-mediated signalling induces its downregulation[25]. This is in line with our data suggesting that IL-7Rα is degraded in an LC3B+ compartment during T cell activation.

IL-7Rα is a subunit of the heterodimeric cytokine receptor IL-7R, and its signalling is crucial for peripheral T cell survival and homeostatic proliferation[26]. It is constantly internalised and recycled through membrane-enveloped organelles and therefore may be found in LC3-containing compartments other than autophagosomes[27,28]. To ensure that IL-7Rα is found in bona fide autophagosomes, we performed immunostaining of IL-7Rα with LC3 or WIPI2, an early autophagosomal marker. We observed co-localisation of IL-7Rα with both molecules when cells are either activated or not (Fig. 3d), and the co-localisation is increased upon BafA1 treatment (Fig. 3e, f). Therefore, we confirmed that IL-7Rα was indeed localised in the autophagosomal compartment.

### IL-7Rα accumulation in T-Atg7$^{-/-}$ CD4+ T cells is due to impaired autophagy

To validate whether lack of autophagy impairs the degradation of IL-7Rα, we measured IL-7Rα expression level in CD4+ T cells freshly isolated from the spleen of T-Atg7$^{-/-}$ mice by flow cytometry. IL-7Rα was upregulated both at the surface and at whole-cell level in naïve T-Atg7$^{-/-}$ CD4+ T cells (Fig. 3g). By contrast, no significant changes in effector T cells were observed. To make sure this is an autophagy-related effect and not an *Atg7*-specific effect, we further confirmed this result by deleting *Atg16l1*, another autophagy gene, specifically in T cells (Supplementary Fig. 3e). The increased expression of IL-7Rα protein was not due to enhanced transcription of *Il7ra* as indicated by the qRT-PCR performed on different days after activation (Supplementary Fig. 3f). It is worth noting that both surface and whole-cell levels of IL-7Rα remained higher in T-Atg7$^{-/-}$ CD4+ T cells after one day of activation, while the difference disappeared on day 3 (Supplementary Fig. 3g).

Previous studies showed that key autophagy genes, like *Atg7* and *Atg5*, control the internalisation of surface receptors[29,30]. To address whether the accumulation of IL-7Rα in autophagy-knockout CD4+ T cells is caused by decreased internalisation, we measured the amount of IL-7Rα binding to the biotin-conjugated antibody at the cell surface over time. We confirmed that the surface level of IL-7Rα was

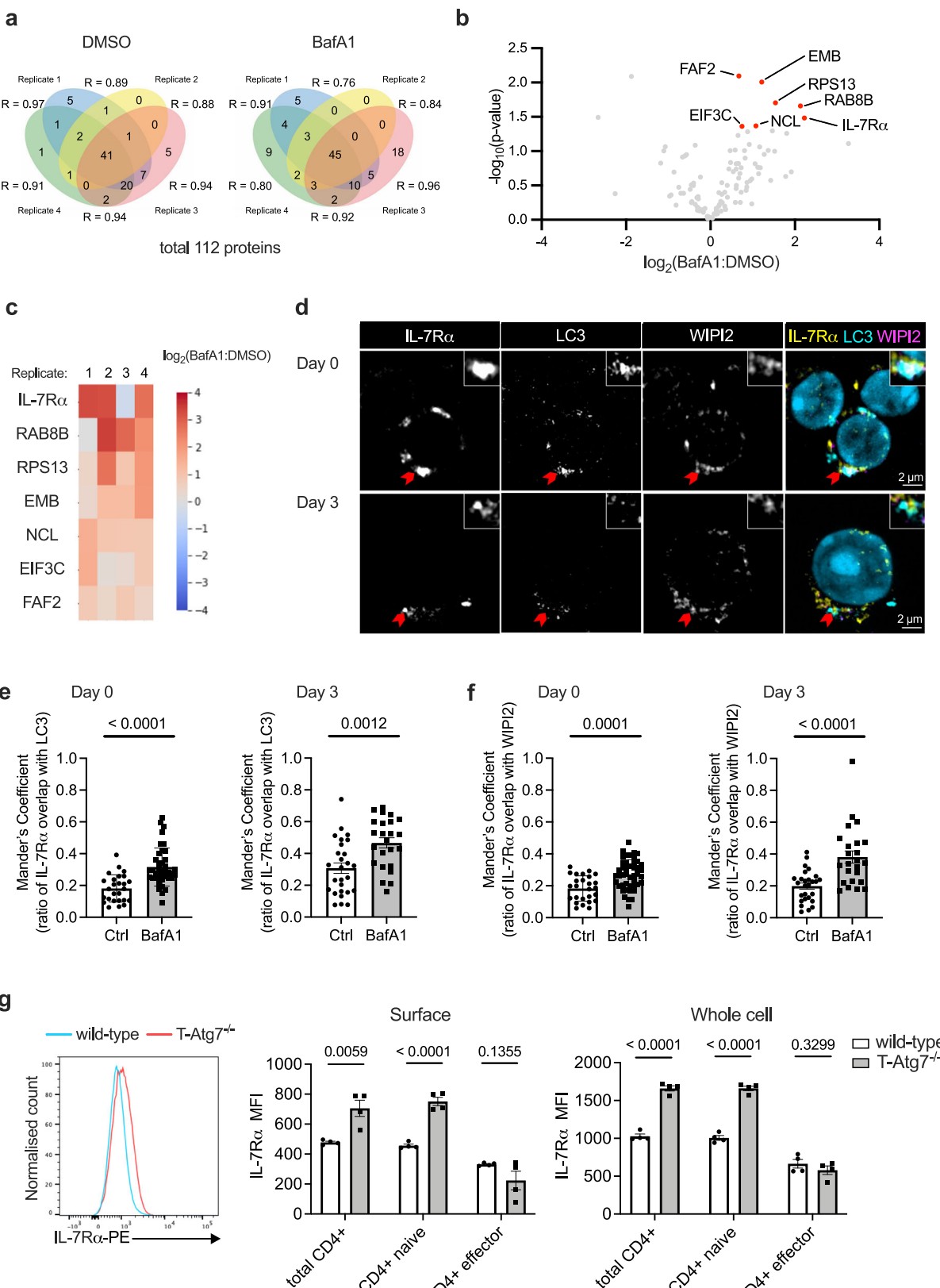

generally higher in T-Atg7$^{-/-}$ naïve CD4+ T cells in comparison to their wild-type counterparts, but could not observe any differences in the rate of receptor internalisation, represented by the percentage of internally trafficked IL-7Rα (Supplementary Fig. 3h). Moreover, surface and intracellular protein levels of IL-7Rα were significantly increased in wild-type naïve CD4+ T cells when treated with SBI-0206965

(Supplementary Fig. 3i), an ULK1/2 inhibitor blocking the initiation phase of autophagy[31], and 3-methyladenine (3-MA; Supplementary Fig. 3j), a PI3K inhibitor blocking the nucleation phase of autophagy[32]. This further confirms that IL-7Rα is degraded through classical autophagy. Therefore, we ruled out the possibility that the surface accumulation of IL-7Rα is due to impaired internalisation.

**Fig. 3 | IL-7Rα is degraded via autophagy in CD4+ T cells. a** Venn diagrams depicting proteinase K (Prot K)-protected, biotinylated proteins identified in activated CD4+ T cells with more than two spectral counts across four biological replicates in both DMSO and bafilomycin A1 (BafA1) treated groups. Pearson's correlation coefficients of the label-free quantity of each protein included in two replicates are displayed. **b** Volcano plot of proteins from **a** labelled by LC3B-AP2 in activated CD4+ T cells. Proteins significantly upregulated in response to BafA1 treatment are highlighted in red, $p < 0.05$ by unpaired two-tailed Student's $t$ test, $n = 4$ biological replicates per group, with each replicate combining T cells from two mice. **c** A $\log_2$(BafA1:DMSO) heat map of the BafA1-upregulated candidates. Red colour indicates an upregulation in BafA1-treated samples, while blue represents a decrease. **d** Confocal imaging of interleukin-7 receptor-α (IL-7Rα), LC3 and WIPI2 in wild-type CD4+ T cells without activation or activated for 3 days. Red

arrows indicate co-localisation events, whose magnifications are displayed at the upper-right corners. These experiments were repeated as biological triplicates with similar results. **e**, **f** Bar graphs show Mander's correlation coefficient between IL-7Rα and WIPI2 (**e**), as well as IL-7Rα and LC3 (**f**), on non-activated wild-type CD4+ T cells (Day 0) or activated for 3 days (Day 3), $n = 26$ cells for Ctrl group on day 0; $n = 40$ cells for BafA1 group on day 0, $n = 27$ cells for Ctrl group on day 3, $n = 24$ cells for BafA1 group on day 3. **g** Histogram (left) of IL-7Rα surface level, gated on splenic CD4+ T cells from wild-type or T-Atg7$^{-/-}$ mice. Bar graphs show the surface (middle) or whole-cell (right) level of IL-7Rα in total CD4+ T cell, naïve CD4+ T cell (CD62L$^+$ CD44$^{lo}$) or effector CD4+ T cell (CD62L$^-$ CD44$^{hi}$), $n = 4$ mice per group. All values are represented as mean ± SEM with unpaired two-tailed Student's $t$ test. Exact $p$ values are depicted in the figure. Quantitative analyses are representative of four independent experiments. Source data are provided as a Source Data file.

## Excessive IL-7Rα on autophagy-deficient T cell surface impairs IL-2R assembly and signalling

While we confirmed that the accumulation of IL-7Rα in CD4+ T cells is autophagy-dependent, the molecular basis of how it leads to impaired proliferation remained elusive. Both IL-7R and IL-2R share a common gamma-chain ($\gamma_c$) to transduce downstream signalling[33]. IL-7R (heterodimer comprised of $\gamma_c$ and IL-7Rα) is highly expressed on naïve T cells and is co-expressed with the low-affinity IL-2R (heterodimer comprised of $\gamma_c$ and IL-2Rβ). During TCR-mediated activation, IL-7Rα is downregulated from the cell surface and IL-2Rα (also known as CD25) takes over to form high-affinity IL-2R (heterotrimer comprised of $\gamma_c$, IL-2Rα and IL-2Rβ). When binding to IL-2, high-affinity IL-2R mediates persistent and robust downstream signalling, which is crucial for the proliferation of CD4+ T cells[34]. It has been reported in regulatory T cells that $\gamma_c$ has limited availability and pre-associates with excessive IL-7Rα independently of IL-7, which impedes IL-2R signalling[35,36]. Therefore, we hypothesised that excessive IL-7Rα expression leads to delayed proliferation of T-Atg7$^{-/-}$ CD4+ T cells by a similar mechanism (Fig. 4a).

To test whether IL-2R signalling is impaired in CD4+ T cells lacking autophagy, we measured the level of phosphorylated Stat5 (pStat5), the classical downstream transducer of IL-2 signalling. As naïve T cells respond poorly to IL-2, we hypothesised that IL-2 signalling contributes to T cell activation mainly after high-affinity IL-2R is expressed[37]. We observed an upregulated expression of IL-2Rα 24 h after TCR-mediated activation (Supplementary Fig. 3a), while IL-7Rα remained accumulated on the cell surface of autophagy-deficient CD4+ T cells at the same time point (Supplementary Fig. 4a). To determine whether accumulated IL-7Rα impacts on IL-2-driven signalling, we incubated activated CD4+ T cells with different doses of IL-2 (Fig. 4b). Corroborating our initial hypothesis, pStat5 level was higher in presence of increased concentrations of IL-2 in wild-type CD4+ T cells, whereas IL-2R signalling in T-Atg7$^{-/-}$ T cells was refractory to the administration of IL-2 (Fig. 4c). Importantly, this was not due to different expression of IL-2R, as we found no significant differences in the expression of its three subunits between T-Atg7$^{-/-}$ and wild-type T cells (Supplementary Fig. 4a) on day 1 post activation. However, due to the impaired IL-2 signalling, the expression levels of IL-2Rα, controlled by IL-2 signalling, were significantly decreased in T-Atg7$^{-/-}$ T cells on day 3 post activation (Supplementary Fig. 4b).

To further confirm whether the assembly of these cytokine receptors is impaired in T-Atg7$^{-/-}$ CD4+ T cells, we measured the co-localisation of $\gamma_c$ with either IL-7Rα or IL-2Rα. Since IL-2 signalling is polarised at the immunological synapse, we incubated the activated T cells on supported lipid bilayers (SLBs) containing ICAM1 and anti-CD3 proteins, in the presence of IL-2[38]. We then visualised the receptor subunits at the immunological synapse (Fig. 4d, left). As expected, we found a significantly lower colocalisation of IL-2Rα and $\gamma_c$ in T-Atg7$^{-/-}$ CD4+ T cells (Fig. 4d, right), indicating that the formation of IL-2R complex was indeed impaired in autophagy-deficient CD4+ T cells at

the immunological synapse. However, we did not see any significant differences in the co-localisation of $\gamma_c$ with IL-7Rα (Supplementary Fig. 4c). To evaluate whether IL-7Rα sequestration happened outside the immunological synapse, we performed confocal analysis of whole cell volumes, where we could observe that $\gamma_c$-IL-7Rα co-localisation is indeed higher in T-Atg7$^{-/-}$ CD4+ T cells (Fig. 4e). Moreover, c-MYC and mTORC1, two direct downstream targets of IL-2 signalling pathway which govern T cell metabolism[39,40], displayed a significant decrease in autophagy-deficient CD4+ T cells during activation (Supplementary Fig. 4d, e). This may also contribute to their delayed proliferation upon antigen stimulation. Taken together, our results suggest that autophagy plays a key role in CD4+ T cell activation, by degrading IL-7Rα, which releases $\gamma_c$ and promotes its assembly in high-affinity IL-2R, a signalling receptor that is essential for T cell proliferation.

## Discussion

In this study, we dissected the molecular mechanism of autophagy in the antigen-induced proliferation of CD4+ T cells. Using a powerful mouse model based on the proximity biotinylation technique, we demonstrate that autophagic degradation of IL-7Rα plays a crucial role in T cell activation, since it can competitively inhibit IL-2R signalling through sequestration of the limited amount of $\gamma_c$. Thus, we identified autophagy's role in the transition of cytokine signalling, from IL-7R-mediated homeostasis in naïve T cells to IL-2R-mediated expansion in effector T cells. This research not only provides proof-of-principle application of the *Lc3b-AP2* mouse model but also fills a key gap in our knowledge of CD4+ T cell activation.

We observed that IL-7Rα levels were higher in autophagy-deficient naïve CD4+ T cells, while the difference disappears on day 3 post-activation. In effector T cells, the expression of IL-7Rα is transcriptionally inhibited, thus stopping the production of new IL-7Rα[41]. Moreover, previous studies indicated that IL-7Rα can be degraded through both lysosome- and proteasome-mediated pathways in naïve T cells[27,28]. It suggests that other compensatory pathways for degradation of excessive IL-7Rα might take place. During T cell activation, genes of proteasome subunits are upregulated, which may contribute to the compensatory degradation of IL-7Rα in autophagy-deficient CD4+ T cells[42]. Other pathways compensating gradually for the loss of autophagy in T-Atg7$^{-/-}$ CD4+ T cells could be the explanation for why the cells eventually catch up on their proliferation in vitro, and why IL-7R is not upregulated in the T-Atg7$^{-/-}$ effector CD4+ T population.

According to iLIR, a web resource that allows the prediction of Atg8-family interacting proteins, the murine IL-7Rα contains a LIR domain in its intracellular domain from amino acid 399 to 404 (PVYQDL). Therefore, it is possible that IL-7Rα is recruited to autophagosomes through direct interaction with LC3, either after its internalisation through the endosomal pathway or on its way to the plasma membrane.

Our observation of accumulated IL-7Rα in both *Atg7-* and *Atg16l1-*deficient CD4+ T cells is in contrast to a report in T cells lacking Vps34,

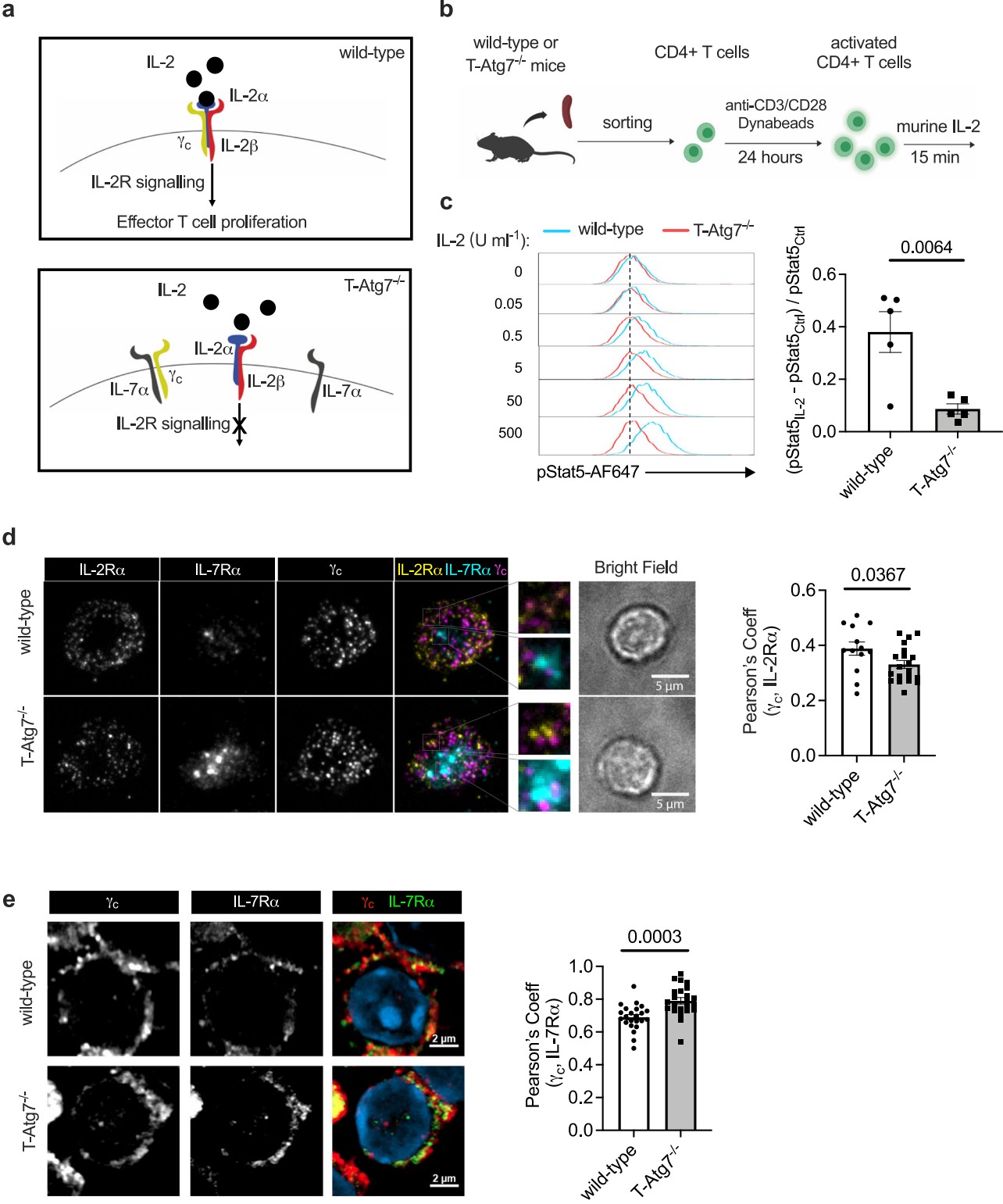

a PI3-kinase involved in the initiation of autophagy[43]. In this study, McLeod et al. describe that Vps34-deficient CD4+ T cells downregulate the surface expression of IL-7Rα. However, the autophagic flux in that model is not impaired and the lower level of surface IL-7Rα was proposed to be due to a deficit in the retromer pathway mediated by Vps34. By contrast, another study in which a bone marrow chimaera model was generated to exclude the effect of lymphopenia, IL-7Rα expression is higher in Vps34-deficient CD4+ T cells compared to autophagy-proficient cells, further corroborates our findings, which were also made in a non-lymphopenic environment[44].

Moreover, autophagy may contribute to the differentiation and memory formation of CD4+ T cells through modifying common gamma chain cytokine signalling (IL-2/IL-15). IL-2 signalling is antagonistic to Th17 differentiation since it suppresses the expression of IL-6Rβ (or gp130)[45,46]. Consistent with a previous study which shows an expansion of IL-17-producing cells in *Lck^{Cre}, Atg7^{flox/flox}* mice[47], the decreased IL-2 signalling in autophagy-deficient CD4+ T cells may facilitate its differentiation into Th17. Also, as Gros et al. showed in their recent study, the formation of memory CD4+ T cells is impaired without autophagy[5]. An IL-2 or IL-15 signalling-dependent mechanism

**Fig. 4 | IL-2R signalling is impaired in autophagy-deficient CD4+ T cells.**
**a** Schematic overview of how excessive interleukin-7 receptor-α (IL-7Rα) may lead
to impaired T cell proliferation. During T cell activation, the interleukin-2 receptor-
α (IL-2Rα) is upregulated on the cell surface, while surface IL-7Rα is decreased. In
T-Atg7$^{-/-}$ CD4+ T cells, excessive IL-7Rα sequestrates the common gamma chain
(γ$_c$), a key subunit of both IL-7R and IL-2R, and abrogates the downstream IL-2R
signalling which is crucial for the expansion of the effector population.
**b** Experimental set-up: naïve CD4+ T cells from either wild-type or T-Atg7$^{-/-}$ mice
were activated with anti-CD3/CD28 Dynabeads for 24 h, followed by treatment with
murine IL-2 for 15 min. Phosphorylated Stat5 (pStat5) level was measured by flow
cytometry. **c** Histograms (left) of pStat5 level in activated CD4+ T cells from wild-
type or T-Atg7$^{-/-}$ mice, stimulated with different doses of murine IL-2. Bar graph
(right) depicts the ratio of increase in p-Stat5 signal in response to 20 ng ml$^{-1}$
murine IL-2 (pStat5$_{IL-2}$ − pStat5$_{ctrl}$) to the basal level (pStat5$_{ctrl}$), determined from

two independent experiments, $n$ = 5 mice per group. **d** TIRF imaging (left) of IL-7Rα,
IL-2Rα and γ$_c$ at the immunological synapse. Naive CD4+ T cells activated for 24 h
were exposed to supported lipid bilayers containing ICAM1 and anti-CD3 proteins,
then fixed after a 10-min incubation with 50 U ml$^{-1}$ murine IL-2. Cytokine receptors
were stained for imaging. Square insets show magnification of the co-localisation
events. Bar graph (right) shows Pearson's correlation coefficient between the IL-
2Rα and γ$_c$, $n$ = 13 cells for wild-type group, $n$ = 20 cells for T-Atg7$^{-/-}$ group.
**e** Confocal imaging of IL-7Rα and γ$_c$ at the surface of activated CD4+ T cells (with
high CD25 expression). Bar graph (right) shows Pearson's correlation coefficient
between the IL-7Rα and γ$_c$ at the cell surface, $n$ = 23 cells per group. All quantitative
analyses are representative of three independent experiments. All data are repre-
sented as mean ± SEM with unpaired two-tailed Student's $t$ test. Exact $p$ values are
depicted in the figure. Source data are provided as a Source Data file.

might also contribute to this phenotype, since IL-2 has been shown to
be essential for CD4+ T-cell memory formation and IL-15 for memory
maintenance[48,49]. Additionally, autophagy has been shown to be more
important in the effector expansion of CD4+ rather than CD8+ T cells,
while in CD8+ T cells, the defects appeared only later during the
effector-to-memory cell transition phase, which also negatively
impacts the secondary anti-viral response to the same pathogen[10,11,50].

It is known that proteins firstly described for their function in
autophagy can perform diverse other functions[51]. Indeed, the autop-
hagy machinery also participates in cellular processes other than
classical macroautophagy, such as endosomal microautophagy (eMA),
and in particular, LC3-associated phagocytosis (LAP) and LC3-
associated endocytosis (LANDO), where lipidated LC3 is expressed
on the cytosolic surface of vesicles[24]. However, any proteins associated
with these vesicles would be excluded from our analysis, as LC3 would
be located towards the cytosol, and our protocol specifically enriches
for autophagosomal luminal proteins. Another pathway to be con-
sidered is the LC3-dependent extracellular vesicle loading and secre-
tion (LDELS), where lipid-conjugated LC3s reside in the lumen of
extracellular vesicles[52]. In fact, using a pulse-chase proximity labelling
strategy, the LC3-dependent secretome of extracellular vesicles was
recently profiled in vitro[52]. Interestingly, we found a putative target
potentially involved in this pathway. Rab8b, a small Rab GTPase
indispensable for exocytic trafficking of post-Golgi vesicles to the
plasma membrane was revealed as a putative cargo in the LC3 com-
partment in activated CD4+ T cells (Fig. 3b, c)[53]. In addition to this
role, the protein has also been suggested to be co-localised with LC3
to mediate autophagy-based secretion and autophagosome
maturation[54,55]. In summary, since membrane-bound LC3 is prevalent
in degradative and non-degradative vesicles, the *Lc3b-AP2* mouse
model is not only a powerful technique to reveal macroautophagic
cargo but may also identify other cargoes when BafA1 is replaced with
different specific inhibitors. In addition, this model will be useful to
explore the cytosolic subproteome of LC3 under physiological con-
ditions when the Prot K treatment is left out.

Two other molecules, CDKN1B/p27Kip, a cell cyclin-dependent
kinase inhibitor, and PTPN1, a protein tyrosine phosphatase, were
shown to be removed by autophagy during the early activation of naïve
CD4+ T cells and Th1 cells respectively[32,56]. CDKN1B/p27Kip1 was found
upregulated by Jia et al. using whole-cell proteomics of autophagy-
knockout T cells[56]. Further validation showed that the autophagy-
mediated degradation of CDKN1B is a prerequisite for the proliferation
of naïve CD4+ T cells. Proteomics and western blots have been widely
used to indirectly map autophagosomal content from autophagy-
deficient T cells. However, these techniques cannot exclude that cells
without autophagy may have synthesised these as new proteins. By
purifying autophagosomes with extensive subfractionation from acti-
vated Th1 cells, Mocholi et al. showed that autophagy degrades
PTPN1[32]. It is possible that we did not identify either of these proteins
due to the duration of activation. Both studies used T cells activated

for no longer than 24 h, while we activated cells for three days to
maximise autophagy levels and cell numbers.

The *Lc3b-AP2* mouse model overcomes the obstacles of previous
methods and facilitates the identification of transient binding partners
and subcellular structures. One caveat of the method is the biased
detection of the proteome. Proteins embedded in large complexes, or
with very few electron-rich amino acid residues exposed at the surface,
have a lower chance to be tagged by the biotin-phenol radicals, which
limits the use of this method to detect certain types of proteins. In our
study, activated primary CD4+ T cells show greater biological and
technical variation than imMEFs. This is most likely due to the high
amount of systemic variation in immune development and variation
induced by activation/ expansion in T cells, which made this pilot study
particularly challenging. In addition, T cells are smaller in size with little
cytoplasm compared to imMEFs. As a result, a large number of T cells
were required to map autophagy-regulated proteins. However, this
pilot study has proven that it is possible to detect abundant proteins
(such as IL-7Rα) targeted for autophagic degradation in T cells. Iden-
tifying less abundant proteins targeted by the autophagosome in
T cells will require further optimisation and more repeats. This is not
necessarily true for other primary cell types. Therefore, our model,
together with other murine models based on this technique, indicates
their broader application for the study of proteins in organelles in
primary cells and in vivo[57-62].

Overall, it is likely that autophagy is required to selectively
degrade a spectrum of molecules at different time points to lift the
brake on T cell proliferation, which is finely tuned by multiple signal-
ling pathways. An autophagic programme will also be key to remo-
delling the cell and providing energy, nutrients and building blocks for
protein synthesis and for the generation of the daughter cells.

## Methods
### Mice
*Atg7$^{flox/flox}$* mice (from M Komatsu) and CD4$^{cre}$ mice (from Adeline Haj-
jar) were crossed to obtain CD4$^{cre}$ *Atg7$^{-/-}$* mice (T-Atg7$^{-/-}$) on a C57BL/6
background. All mice were 6–8 weeks of age at the start of each
experiment and were matched in age. Both genders are included
CD4$^{cre}$ *Atg7$^{+/+}$* littermates were used as wild-type controls (wild-type).
C57BL/6 SJL CD45.1 mice for bone marrow chimaera and T cell adop-
tive transfer were purchased from Charles River, UK. OT-II mice were
crossed with CD4$^{cre}$ *Atg7$^{-/-}$* mice. CD4$^{cre}$ *Atg16l1$^{-/-}$* mice were from Kevin
Maloy. *Lc3b-APEX2* (*Lc3b-AP2*) mice were generated by Cyagen, by
constitutively knocking in an APEX-tag and a GGGGSGGGGGS-linker
into the exon 1 of the mouse *Map1lc3b* allele.

All mice were held in Biological Support Unit at the Kennedy
Institute of Rheumatology. All mice were under specific pathogen-free
level maintenance at 24 °C, 50% humidity and a 12:12 h light/dark cycle.
Mice were kept in individually ventilated cages, with ad libitum access
to autoclaved water and irradiated food pellets. Cages were changed
once weekly, and the health status of the mice was monitored based on

body weight, coat and behaviour. Littermates from both sexes were randomly assigned to experimental groups. The exact number of the mice used for the experiments are indicated for each experiment in the figure legends. Animal experiments were performed in accordance with institutional policies, British federal regulations, and were approved by the UK Home Office (project license PPL 30/3388).

### Genotyping

The identification of the mouse genotypes was performed on mouse ear clips. The ear tissue was lysed with 200 μl 50 mM NaOH at 95 °C for 30 min. After adding 25 μl 1 M Tris-base buffer (pH = 8) for neutrolisation, vortex the mixture and centrifuge 5 min at 16,000×$g$ at 4 °C (max speed). The supernatant was used for conducting PCR or stored at −20 °C.

The PCR reactions were performed with MyTaq Red Mix (Applied Biosystems, USA). Primers for the *Lc3b-APEX2*, CD4$^{cre}$, *Atg7$^{flox}$*, and *Atg16l1$^{flox}$* transgenes were listed in Supplementary Table 1.

### Viral vectors and lentivirus production

P-Lenti CMV/TO SV40 small + Large T (w612-1) was a gift from Eric Campeau (Addgene plasmid # 22298). For packaging the virus, HEK293T cells were transfected with vector and replication-incompetent lentiviral packaging constructs, harvested after 48-hour post-transfection, and viral supernatants filtered through a 0.45 μM cellulose acetate filter.

### Cell line generation

The mouse embryonic fibroblasts (MEFs) were generated according to published protocols[63]. In brief, *Lc3b-AP2* homozygous embryos were separated from female uteri on E13.5. After removing the head above eyes and red organs (heart and liver), the rest of the embryos were minced and subjected to 0.25% Trypsin (Sigma) digestion, with intermittent pipetting up and down. Digestion was terminated by MEF culture media (Dulbecco's Modified Eagle Medium, Sigma-Aldrich) supplemented with 10% Fetal Bovine Serum (Sigma-Aldrich), 2% penicillin-streptomycin (Sigma-Aldrich) and 2 mM L-Glutamine (Sigma-Aldrich). Single cells and clusters were transferred onto the T25 flask and incubated with MEF culture media overnight. After replacing the old media and removing the debris, cells were passaged until reaching confluency. To immortalise the *Lc3b-AP2* cell line, MEFs were re-seeded to 6-well plates and transduced with the supernatant of the lentivirus expressing SV40 small and large T antigens overnight. Monoclonal immortalised MEFs were selected through dilution cloning and over 10 rounds of passaging.

### Bone marrow chimaera

Bone marrow (BM) chimaeras were generated similarly as described before[10]. After erythrocyte lysis, BM cells extracted from a single 8-week-old wild-type or T-Atg7$^{-/-}$ (both CD45.2) were 1:1 mixed with C57BL/6 SJL mouse (CD45.1) in a total volume of 200 μl PBS, with 3 × 10$^6$ cells each. The mixture was injected intravenously into C57BL/6 SJL CD45.1 recipients 2 h after being lethally irradiated (450 cGy twice, 4 h apart). Followed by an 8-week reconstitution, BM chimaeras were immunised with 1 × 10$^6$ PFU MCMV (Smith strain ATCC:VR194).

### T-cell adoptive transfer

For adoptive transfer, 2 × 10$^4$ purified splenic OT-II T cells from CD45.2 wild-type or T-Atg7$^{-/-}$ mice were intravenously injected into C57BL/6 SJL CD45.1 recipients. After 24 h, recipient mice were immunised with 1 × 10$^6$ IU recombinant adenovirus H5-ovalbumin (abm).

### CD4+ T Cell isolation

Total CD4+ T cells were purified with EasySep Mouse T Cell Isolation Kit (Stemcell Technology). Total splenocytes suspended in MACS buffer (2% FBS in DPBS) were incubated with rat serum and isolation cocktail for 10 min at room temperature, then incubated with Streptavidin RapidSpheres for 2.5 min. After topped up to 2.5 ml, tubes were placed into the magnet and incubated at room temperature for 2.5 min. Purified CD4+ T cells were transferred to new tubes for further analysis.

### CD4+ T cell in vitro activation

T cells were cultured in RPMI-1640 Medium (Sigma-Aldrich) containing 10% Fetal Bovine Serum (Sigma-Aldrich), 2% penicillin-streptomycin (Sigma-Aldrich) and 2 mM L-Glutamine (Sigma-Aldrich). For antigen-induced activation, total splenocytes isolated from OT-II mouse spleens were cultured with OVA 323−339 peptide (1 mg/ml, InvivoGen) and recombinant murine IL-2 (20 ng/ml, PeproTech). For anti-CD3/CD28-induced activation, CD4+ T cells were stimulated with Dynabeads Mouse T-Activator CD3/CD28 (Gibco) and recombinant murine IL-2 (20 ng/ml, PeproTech), following manufacturer's instructions.

### Quantitative PCR analysis

Total RNA isolated using an RNeasy Plus Mini Kit (Qiagen, 74134) underwent reverse transcription with the High Capacity RNA-to-cDNA Kit (Thermo Fisher, 4387406). Predegisned Taqman probes (Thermo Fisher, *Atg7*: Mm00512209_m1; *Il7ra*: Mm00434295_m1; *Gapdh*: Mm99999915_g1), TaqMan Gene Expression Master Mix (4369016, Thermo Fisher) and the ViiA 7 Real-Time PCR System (Thermo Fisher) were used for quantitative PCR. ΔΔCt method was used for the quantification of target mRNAs expression using *Gapdh* as the reference gene.

### Flow cytometry

The following antibodies were used for flow cytometry (dilution in brackets) – from BioLegend: anti-CD4 BV605 GK1.5 (1:400), anti-CD3 AF700 17A2 (1:200), anti-CD25 PerCP-Cy5.5 PC61 (1:200), anti-CD62L APC-Cy7 MEL-14 (1:400), anti-CD69 PE-Cy7 H1.2F3 (1:400), anti-CD127 (IL-7Rα) PE A7R34 (1:200 for surface staining, 1:100 for intracellular staining), anti-CD45.2 FITC 104 (1:100); from BD Biosciences: anti-CD122 (IL-2Rβ) BV650 5H4 (1:200), anti-CD132 (γ$_c$) BV650 TUGm2 (1:200); from eBioscience: anti-TCRVα2 APC B20.1 (1:400), anti-CD44 PE-Cy7 IM7 (1:200), anti-CD8a FITC 53-6.7 (1:400), anti-phospho-S6 (Ser235, S236) APC cupk43k (1:50); from Cell Signalling: anti-phospho-Stat5 (Tyr694) AF647 C71E5 (1:100), anti-c-Myc D84C12 (1:200); from Luminex: anti-LC3 FITC 4E12 (1:20).

To assess the surface receptor level, cells were washed with ice-cold phosphate buffer saline (PBS) and incubated for 20 min at 4 °C in the dark, with cold PBS containing Fc block (0.125 g/ml Biolegend) and antibodies. After washing with cold PBS, cells were fixed for 10 min at room temperature with 2% Paraformaldehyde and analysed using LSRFortessa™ X-20 Cell Analyzer (BD Biosciences), LSR II Flow Cytometer (BD Biosciences) and Flowjo software (Tree Star). For intracellular staining of IL-7Rα, cells were fixed with Fixation Buffer (BioLegend) in the dark for 20 min at room temperature after surface staining, then permeabilised with Intracellular Staining Perm Wash Buffer (BioLegend) and incubated with anti-IL-7Rα antibody 1:100 diluted in Perm/Wash Buffer in the dark for 1 h at room temperature. Cells were extensively washed with Perm/Wash buffer and resuspended with FACS buffer for further flow cytometry analysis.

To measure cell proliferation profile in vitro, cells isolated from mouse spleens were stained with CellTrace Violet (5 μM, Invitrogen) for 20 min at room temperature in dark. After extensive washing, cells were counted and seeded for activation.

To identify viable cells, cells were stained with LIVE/DEAD Fixable Aqua Stain Kit (1:400, Invitrogen) or LIVE/DEAD Fixable Near-IR Stain Kit (1:1,000, Invitrogen) diluted with PBS at 4 °C for 20 min.

Tetramer staining was performed similarly as previously described[10]. In brief, before surface staining, cells were incubated with 20 μg/ml tetramer in PBS at 37 °C for 30 min. Peptide sequences for

MCMV tetramers were as follows: m45 [985]HGIRNASFI[993], H-2Db-restricted; m38 [316]SSPPMFRV[323], H-2Kb-restricted; IE3 [416]RALEYKNL[423], H-2Kb-restricted. MHC-Class I monomers were stored at −80 °C. Tetramers were made with biotinylated monomers with PE- or APC-conjugated Streptavidin to achieve 1:1 ratio with biotin binding sites, then added in 1/10th volumes waiting 10 min between additions. Tetramerized complexes were then stored at 4 °C.

To assess the phospho-Stat5 level, cells were fixed with pre-heated 2% PFA for 15 min at 37 °C after the live-dead staining, permeabilised with pre-cooled 90% methanol for more than 4 h and stained overnight with the phospho-Stat5 antibody diluted with 0.5% BSA in PBS at 4 °C.

To measure the autophagic flux, cells were treated with bafilomycin A1 (10 nM) or DMSO for 2 h before staining. We adapted the Guava Autophagy LC3 Antibody-based Detection Kit (Luminex) as follows: After surface staining, cells were washed once with Assay Buffer. After permeabilization with 0.05% Saponin, cells were spun immediately and incubated with anti-LC3 (FITC) antibody (1:20 diluted in Assay Buffer) at room temperature for 30 min in dark. After extensive washing with Assay Buffer, cells were fixed with 2% PFA and underwent flow cytometry analysis. Autophagic flux was calculated as LC3-II mean fluorescence intensity of (BafA1-Vehicle)/Vehicle.

**IL-7Rα internalisation assay**
IL-7Rα internalisation assay was performed similar as described before[30]. In brief, CD4+ T cells were purified from the spleens of wild-type or T-Atg7$^{-/-}$ mice. After incubation with biotin-conjugated anti-IL-7Rα antibody (BioLegend, A7R34) 1:50 diluted in DPBS on ice for 30 min, cells were extensively washed and fractionated into the following groups ($5 \times 10^6$ cells/group): 90 min on ice, 90, 60 and 30 min at 37 °C, 5% $CO_2$. After incubation, cells were stained with AF647-Streptavidin (BioLegend, 1:1000), anti-CD4 BV605 GK1.5 (BioLegend, 1:400), anti-CD44 BV785 IM7 (BioLegend, 1:400), LIVE/DEAD Fixable Aqua (Invitrogen) stain for 20 min on ice and fixed with 2% PFA. The % internalised IL-7Rα is calculated as the Eq. (1) (MFI: mean fluorescence intensity):

$$\%\text{internalised IL-7R}\alpha = 100 - \left[ \left( \frac{\text{MFI sample}}{\text{MFI 90 minice}} \right) \times 100 \right] \quad (1)$$

**Western blots**
Cells were washed with cold PBS and lysed using RIPA lysis buffer (Sigma-Aldrich) supplemented with complete Protease Inhibitor Cocktail (Roche) and PhosSTOP (Roche). Total protein concentration in supernatant after spinning down the debris was quantified by BCA Assay (Thermo Fisher). Samples were added with 4x Laemmli Sample Buffer (Bio-Rad) and boiled at 100 °C for 5 min, with 20–30 μg protein per sample for SDS-PAGE analysis. 4–20% Mini-PROTEAN TGX Precast Protein Gels (Bio-Rad) with Tris/Glycine/SDS running buffer (Bio-Rad) was used. PVDF membrane (Millipore) transferred with protein were blocked with 5% skimmed milk-TBST. Membranes were incubated overnight with the following primary antibodies diluted in 1% skimmed milk: LC3B (2775, Cell Signalling, 1:1000), APEX (IgG2A) (custom made, Regina Feederle, 1:200), GAPDH (MAB374, Millipore, 1:10,000), β-actin (3700, Cell Signaling, 1:5,000), ATG9A (ab108338, Abcam, 1:1,000), followed by 1 h room incubation at room temperature with secondary antibodies diluted in 1% milk with 0.01% SDS: IRDye 680LT Goat anti-Mouse IgG (H + L) (Licor, 1:7,500), IRDye 680LT Goat anti-Rat IgG (H + L) (Licor, 1:10,000) and IRDye 800CW Goat anti-Rabbit IgG (H + L) (Licor, 1:10,000). Biotinylated proteins were detected by IRDye 680RD Streptavidin (Licor, 1:1000) diluted in 1% milk with 0.01% SDS. Membranes were imaged with the Odyssey CLx Imaging System (Licor). Data were analysed using Image Studio Lite (Licor).

**Immunofluorescence staining and confocal microscopy**
For intracellular staining of imMEFs, cells were seeded on coverslips in tissue culture plates. After reaching 70% confluency, proximity-labelled imMEFs were washed with ice-cold PBS and fixed with 2% paraformaldehyde (Invitrogen) for 10 min, then permeabilised with 0.5% Triton X for 15 min and blocked with PBS containing 0.5% bovine serum albumin (Sigma-Aldrich) for 1 h at room temperature.

For intracellular staining of purified CD4+ T lymphocytes, cells were washed in PBS and transferred on PolyL-Lysine (Sigma-Aldrich) treated coverslips, followed by incubation for 30 min at 37 °C. For intracellular staining, cells were fixed with 2% paraformaldehyde (Sigma-Aldrich) for 10 min and permeabilised with 0.1% Triton X (Sigma-Aldrich) for 10 min, then blocked in PBS containing 2% bovine serum albumin (Sigma-Aldrich) and 0.01% Tween 20 (Sigma-Aldrich) for 1 h at room temperature.

For surface staining of purified CD4+ T lymphocytes, activated cells were washed in PBS and transferred on PolyL-Lysine (Sigma-Aldrich) treated coverslips, followed by incubation for 30 min at 37 °C. After cytokine treatment for 15 min, cells were washed and stained with antibodies at 4 °C for 20 min and fixed with 2% PFA.

All antibodies for immunofluorescence stain are as follows: FITC-LAMP1 (121605, BioLegend, 1:100), WIPI2 (SAB4200399, Sigma, manually conjugated with AF568, 1:100), AF647-Streptavidin (405237, BioLegend, 1:1,000), LC3 (PM036, MBL International, manually conjugated with AF488, 1:100), APEX2 (IgG2A, custom made from Regina Feederle, manually conjugated with AF568, 1:200), IL-7Rα (135002, BioLegend, manually conjugated with AF488, 1:100), IL-2Rα (154202, BioLegend, manually conjugated with AF568, 1:100), γ$_c$ (132307, BioLegend, manually conjugated with AF647, 1:100).

After staining with antibodies for 45 min at room temperature and washed, samples were stained with DAPI (1 μg/ml, Thermo Fisher), mounted to slides with SlowFade Gold Antifade Mountant (Invitrogen) and imaged with Zeiss LSM 980 confocal microscope 63x oil immersion-lens (Zeiss). Data were analysed with ZEN Blue (Zeiss) and ImageJ software. Pearson correlation coefficients were obtained using JACoP pug-in[64].

**Supported lipid bilayer preparation and use**
Glass coverslips were plasma cleaned and mounted onto six-channel chambers (Ibidi). SLB were prepared as previously described[65], using 1,2-dioleoyl-sn-glycero-3-[(N-(5-amino-1-carboxypentyl) iminodiacetic acid) succinyl]-Ni (NTA-lipids, Avanti Polar Lipids), biotinylated 1,2-dioleoyl-sn-glycero-3-phosphoethanolamine (biotinyl cap PE-lipids, Avanti Polar Lipids) and 1,2-dioleoyl-sn-glycero-3-phosphocholine (DOPC-lipids, Avanti Polar Lipids). Channels in Ibidi chamber were covered with liposome mixture and after a 20 min incubation at room temperature, they were washed and blocked with 3% Bovine Serum Albumin (BSA, Sigma-Aldrich) and 100 μM NiSO$_4$ for 20 min. After incubation with 10 μg/ml streptavidin, bilayers were incubated with mono-biotinylated 2c11 (anti-mouse CD3 antibody) and his-tagged mouse ICAM1-AF405, to reach a molecular density of 200 molecules/μm$^2$. Channels were washed and cells were added on the bilayer and incubated at 37 °C for 10 min before fixation with 4% PFA. Next, samples were permeabilized with 0.1% Saponin for 10 min and blocked with 100 mM Gly 3% BSA for 30 min. Finally, samples were incubated with primary antibodies in 1% BSA.

**Total internal reflection fluorescence microscopy and image analysis**
Imaging was performed on an Olympus IX83 inverted microscope equipped with a TIRF module. The instrument was equipped with an Olympus UApoN 150 × 1.45 NA objective, 4-line illumination system (405, 488, 561 and 640 nm laser) and Photomertrics Evolve delta EMCCD camera. Image analysis and visualisation was performed using

ImageJ software. Pearson correlation coefficients were obtained using JACoP plug-in[64].

### Electron microscopy

*Lc3b-AP2*-expressing imMEFs were grown on aclar sheets (Science Services), supplemented with 4.83 μM Hemin chloride (ROTH) for 16 h and treated with 200 nM BafA1 (Biomol) for 2 h before fixation. Cells were fixed in 2.5% glutaraldehyde (EM-grade, Science Services) in 0.1 M sodium cacodylate buffer (pH 7.4; CB) for 30 min. Fixation and the following processing steps were carried out on ice. After washes in CB, endogenous peroxidases were blocked in 20 mM glycine (Sigma) in CB for 5 min and cells washed in CB. 1x diaminobenzidine (DAB) in CB with 2 mM calcium chloride was prepared from a 10x DAB stock (Sigma) in hydrochloric acid (Sigma) and added to the cells for 5 min without and for another 40 min with 10 mM $H_2O_2$ (Sigma). After washes in CB, cells were postfixed in reduced osmium (1.15% osmium tetroxide, Science Services; 1.5% potassium ferricyanide, Sigma) for 30 min, washed in CB and water and incubated over-night in 0.5% aqueous uranylacetate (ScienceServices). Dehydration was accomplished using a graded series of ice-cold ethanol. Cell monolayers were infiltrated in epon (Serva) and cured for 48 h at 60 °C. Cells were ultrathin sectioned at 50 nm on formvar-coated copper grids (Plano). TEM images were acquired on a JEM 1400plus (JEOL) using the TEMCenter and Shotmeister software packages (JEOL) and analysed in Fiji.

### Proximity labelling

The proximity labelling based on AP2 peroxidase was performed as described before[9]. After 30 min biotin-phenol (500 μM, Iris Biotech) treatment at 37 °C, cells were supplemented with $H_2O_2$ (1 mM, Sigma-Aldrich) for 1 min and immediately quenched three times with cold quenching solution (10 mM sodium ascorbate, 5 mM Trolox and 1 mM sodium azide in DPBS). After 3 times more wash with PBS, the dry pellets of the harvested cells were flash frozen and stored at −80 °C.

### Proteinase K digestion

All steps were performed as previously described at 4 °C or on ice unless specified[66]. The pellets of labelled cells were washed and resuspended in Homogenisation Buffer I (10 mM KCl, 1.5 mM $MgCl_2$, 10 mM HEPES-KOH and 1 mM DTT adjusted to pH 7.5). After incubating on overhead shaker for 20 min, cell suspension was homogenised with a tight-fitting pestle in Dounce homogeniser (Scientific Laboratory Supplies) with 70 strikes and balances with 1/5 volume of Homogenisation Buffer II (375 mM KCl, 22.5 mM $MgCl_2$, 220 mM HEPES-KOH and 0.5 mM DTT adjusted to pH 7.5). After centrifugation with 600 x g for 10 min, clear supernatant containing autophagosomes were transferred to new tubes. For Proteinase K protection assay, supernatant was equally divided into four portions, added with 0.2% Triton X-100 (Merck) and/or 30 μg/mL proteinase K (Roche), or nothing. Following 30 min incubation at 37 °C, PMSF (10 mM, Sigma-Aldrich) was administered to inhibit the proteinase K activity. For mass spectrometry, samples were subjected to 100 μg/mL proteinase K digestion for 1 h at 37 °C followed by PMSF treatment. For control samples 0.1% RAPIGest™ was additionally added to the proteinase K digest. Membrane-protected material was enriched by centrifugation at 17,000x g for 15 min.

### Streptavidin-pulldown and on-beads peptide digestion

The pellets of membrane-protected material were lysed with RIPA buffer containing quenchers (50 mM Tris, 150 mM NaCl, 0.1% SDS, 1% Triton X-100, 0.5% sodium deoxycholate, 1x cOmplete Protease Inhibitor Cocktail (Roche), 1x PhosSTOP (Roche), 10 mM sodium ascorbate, 1 mM Trolox and 1 mM sodium azide), sonicated and centrifuged at 10,000×g for 10 min. The supernatant was incubated with Streptavidin-agarose (Sigma-Aldrich) overnight, which was balanced

with RIPA buffer containing quenchers. After 3x wash with RIPA buffer and 3x wash with 3 M Urea dissolved in 50 mM $NH_4HCO_3$, beads were incubated TCEP (5 mM, Sigma-Aldrich) at 55 °C for 30 min and shaken at 1000×rpm. Samples were alkylated with IAA (10 mM, Sigma-Aldrich) at room temperature for 20 min and shaken at 1000x rpm, further quenched by DTT (20 mM, Sigma-Aldrich) and washed 2x with 2 M Urea dissolved in 50 mM $NH_4HCO_3$. After overnight incubation with trypsin (1 μg/20 μl beads, Promega), supernatants were collected, plus 2x washes with 2 M Urea buffer. The samples were acidified with tri-fluoroacetic acid (1%) and underwent vacuum centrifugation to decrease the volume. After being desalted on C18 stage tips (Thermo Scientific), peptides were reconstituted with 0.5% acetic acid for mass spectrometry analysis.

### Mass spectrometry data analysis

Proteomics data have been generated using a Q-Exactive and Orbitrap Fusion Lumos (both Thermo Fisher). Detailed methodology is available with the mass spectrometry proteomics data as deposited in the ProteomeXchange Consortium via the PRIDE[67] partner repository with the dataset identifier PXD033733. MaxQuant (Andromeda)[68] (version 1.6.10.43) were used for peak detection and quantification of proteins based on RAW data. MS spectra were referred to the manually anno-tated UniProt Mus musculus proteome (retrieved 30/03/2020), using the parameters as follows: full tryptic specificity, allowing two missed cleavage sites, modifications included carbamidomethyl (C) and the variable modification to acetylation (protein N terminus) and oxida-tion (M), and filtered with a false discovery rate (FDR) of 0.01. Analysis of label-free quantification intensities of proteins were log2-transformed with Python programming (Version 3.7.6). Missing values were replaced by random values from a distribution of a quarter of the width, and −1.8 units of the original sample distributions. Pro-teins without greater-than-background values in both replicates for at least one condition were removed. Volcano plots were generated using GraphPad Prism software (Version 8.2.1). The $log_2$(BafA1:DMSO) fold change of each protein is plotted on x versus the $log_{10}$(p-value) of each protein plotted on y. Gene ontology analysis was performed with g:Profiler.

### Statistics and reproducibility

Data were displayed as mean ± SEM in bar graphs. *p* values were determined by two-tailed Student's *t* test with GraphPad Prism soft-ware (Version 8.2.1), with *p* values displayed in the figures. A *p* value < 0.05 was considered significant. Sample size was determined by the authors based on power calculations. The exact n numbers used in each experiment are indicated in the figure legends. Each experi-ment was repeated at least twice independently with similar results. Regarding data exclusion, statistical analyses of confocal microscopy exclude apoptotic and dead cells with abnormal nuclear staining as pre-established. Regarding randomisation, recipient mice were ran-domly assigned to groups at the start of each experiment. In experi-ments using cells from mice, both sexes of littermates with different genotypes were randomly assigned to experimental and control groups. In experiments using cell lines, different plates of cells with same confluency were randomly treated with drugs or vehicle. Blind analysis was carried out during the software analysis step in all experiments, including flow cytometry, immunofluorescence, quanti-tative PCR, proteomics and western blotting. Mouse injections were done blindly by the animal facility staff with no knowledge of geno-types or phenotypes.

For meta-analysis, proteins that were detected in both MS studies were meta-analysed using a *Z*-score meta-analysis weighted by the square root of sample size (*n* = 8 for the first experiment, *n* = 6 for the second), and the Benjamini-Hochberg procedure was used to identify significantly differential abundant proteins with FDR < 0.05.

**Reporting summary**

Further information on research design is available in the Nature Research Reporting Summary linked to this article.

## Data availability

The mass spectrometry data generated in this study have been deposited in the Proteomics Identifications database under accession code PXD033733. UniProt Mus musculus proteome was downloaded from Uniprot database retrieved on 30/03/2020 [https://www.uniprot.org/proteomes/UP000000589]. The raw numbers for charts and graphs are available in the Source Data file whenever possible. Source data are provided within this manuscript. Source data are provided with this paper.

## Code availability

Source code[69] for analysing mass spectrometry data is available at: https://github.com/dzhou93/proximity_labelling_pipeline/commit/2e825476556087ae0cff51310556adb278a83d77. The repository is public and open to collaborative continuous development.

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

## Acknowledgements

We thank Jonathan Webber for assistance in flow cytometry and cell sorting, Ryan Beveridge in producing lentivirus, Helena Coker for assistance in confocal microscopy, Najmeeyah Brown, Patricia Moreira and Daniel Andrew for assistance in mouse colony management. We thank Michael L. Dustin, Mark Coles, Ricardo Fernandes, Raymond Moniz and Anish Suri for scientific discussions. This work was supported by grants from the Wellcome Trust Investigator award 103830/Z/14/Z and 220784/Z/20/Z to A.K.S., Versus Arthritis grants 22617 to G.A., Deutsche Forschungsgemeinschaft (DFG, German Research Foundation) within the frameworks of the Munich Cluster for Systems Neurology (EXC 2145 SyNergy – ID 390857198) and the Collaborative Research Center 1177 (ID 259130777) to C.B., the Wellcome Trust Investigator award 208750/Z/17/Z to L.J., the Kennedy-Chinese Scholarship Council PhD studentship to D.Z., the European Union's Horizon 2020 (under the Marie Sklodowska-Curie grant agreement number 893676) to M.B., the Cue Biopharma Post-doctoral Fellowship, KTRR Cell Dynamics Platform and Wellcome 100262Z/12/Z to J.C. For the purpose of Open Access, the author has applied a CC BY public copyright licence to any *Author Accepted Manuscript* version arising from this submission.

## Author contributions

D.Z., G.A., C.B., D.P., and A.K.S. conceptualised the study. D.Z., S.Z., C.B., R.F., L.J., and A.K.S. devised the methodology. D.Z., G.A., M.B., S.Z., J.C., D.P. S.S., M.S., S.S.H., and X.G. carried out the experiments. D.Z., G.A., J.C., S.S.H., L.J., and A.K.S. analysed and/or interpreted the experimental data. D.Z. and A.K.S. wrote the original draft. M.B., G.A., J.C., C.B., and A.K.S. reviewed and edited the manuscript. L.J., C.B., G.A., and A.K.S. provided the supervision.

## Competing interests

The authors declare no competing interests.
