## [Peer Review File · Nature Communications]

Mapping autophagosome contents identifies interleukin-7 receptor- α as a key cargo modulating CD4+ T cell proliferationREVIEWER COMMENTS

Reviewer #1 (Remarks to the Author):

Zhou et al. have developed an innovative mouse model introducing APEX2 into the LC3b locus, in order to employ proximity labelling and proteomics to detect autophagosomal substrates from primary cells ex vivo. The authors successfully employ this technique in activated CD4⁺ T lymphocytes and identify IL7Ra as the most prominent target of the autophagy pathway in activated CD4⁺ T cells. In support, genetic deletion of Atg7 is found to increase cell surface expression of IL7Ra on the cell surface of naïve but not effect CD4⁺ cells. The authors propose that this increase in IL7Ra results in impaired T cell activation in Atg7-deficient cells by titrating the gamma chain away from IL2R complex. Overall, the studies are well done and support the conclusions that have been drawn. The APEX-LC3B mouse will be an useful resource to the autophagy field. Some additional mechanistic insight into how IL7Ra is being targeted for autophagic degradation in CD4 cells will greatly strengthen the manuscript. Since IL7Ra is located at the cell surface or traffics through the endosomal recycling pathway, it is not entirely clear how such a protein would be captured into autophagosomes.

1) Although the identification of IL7Ra via APEX2-LC3B proximity labelling implicates it as a autophagosome cargo, it is largely unclear how this receptor is being targeted for degradation via the autophagy pathway. Is the degradation of IL7Ra dependent on a LIR-based interaction?

2) Alternatively, are any of the known autophagy cargo receptors functionally required for IL7Ra degradation? It is interesting that Figure 3 suggests that in contrast to MEFs, cargo receptors such as p62 and TAX1BP1 are not enriched in the autophagosomes of CD4⁺ T cells?

3) In Figure 3e, why is there no difference in IL7Ra protein levels between Atg7KO and control CD4 effector cells? Although the overall reduction in IL7Ra compared to naïve CD4 cells makes sense, it is not clear why autophagy no longer involved in the turnover of IL7R in CD4 effector cells.

4) In addition to ATG7, the authors observe similar phenotypes in ATG16KO cells. Both of these ATGs are part of the LC3 conjugation pathway, which mediates noncanonical LAP-like pathways such as LANDO, so it is not entirely clear from the results whether IL7Ra is being targeted via classical autophagy. The authors should functionally evaluate IL7Ra upon loss of an ATG involved in a distinct step of the classical autophagy trafficking process, such as ULK1/2 or ATG14.

Reviewer #2 (Remarks to the Author):

This is a very interesting study where the authors not only present evidence of the generation of a new mouse model that can become a very powerful tool to identify cargo degraded by autophagy in different cell types under different situations using a much easier and in many occasions better method than classical autophagosome isolation, but it also uses that tool to identify IL7R degradation as an important process that may explain how autophagy is able to regulate T cell proliferation and differentiation. The model has been carefully evaluated and optimized to permit the identification of autophagosome cargo minimizing background due to interaction that may occur outside of the autophagosome where LC3 can also be found. The data that supports that IL-7 receptor is degraded by autophagy in activated T cells and may interfere with IL-2 signaling is also conclusive and, as mentioned above, the analysis of all the data obtained from the proteomic analysis may even open the door for further identification of autophagosome cargo in T cells and expand our current knowledge of how autophagy regulates T cells.

I just would like to point a few minor issues that I believe, if addressed, my help strengthen some of the conclusions and get a better glimpse of the implications of this study for T cell biology. When the authors show the experiments that have performed as art of their goal to validate the new mouse model it could be important that they would offer a possible explanation as to why only partial protection and not almost total preservation of p62(easier to understand) and AP2 (maybe needs

some explanation) occurs in cells treated only with PK vs PK +Tx

Also in this regard, a quantification similar to the one they offer in some of their other colocalization studies, could be offered for Fig. 2, just to get a more accurate idea of how many of the biotinylated proteins do actually colocalize with AP2 in the autophagosome or LMAP1 in the autolysosome.

The BM chimeras experiment offer new data that support a central role of autophagy in the CD4+ responses against infection. It is a pity, though, that using MHC-II tetramers in those experiments we cannot know how those effects on CD4+ T cells may also extend to the antiviral CD8+ T cell response in vivo. Did the authors collected any data in their FACS analyses that may shed some light into that? This may be important based on previous reported data that appears to suggest that acute expansion on CD8+ T cells following viral infection may not be modulated by autophagy in vivo, suggesting that autophagy may be more important to CD4+ than CD8+ T cell expansion in vivo.

There are a couple of implications that I think may be quite exciting to expand upon and the authors may consider including them in their discussion. First, it would be the implications that this mechanism may have on T cell differentiation, especially in two specific aspects. First, in line with data the authors had already published, do they think that persistence of IL7R may result, as they show here, in reduced IL2R and IL15R signaling that may account, at least in part, for the defects in the efficient generation of memory they recently reported in a very elegant study. Second, there may be an effect of autophagy on CD4+ T cell effector differentiation based on how it can regulate IL7R levels. One can envision for example a situation where loss of autophagy resulting in reduced common gamma chain receptor would inhibit processes such as Th2 differentiation that rely on signaling from two common gamma chain receptors, IL2R and IL4R; while in other processes such as Th17 differentiation we may see an opposite effect as the negative effect of IL2R signaling would be reduced, while there would be no effect on signaling from other cytokine receptors (not members of the common gamma chain family) that direct Th17 generation.

Finally, it may be interesting if the authors could analyze the IL7Ra sequence to see if it has putative LIRs, that would support its interaction with LC3, and where they are. Though their absence would not preclude the conclusions reached here at all (it could all be autophagy receptor mediated) if present, it may help understand how this process occurs. In that sense I wanted to ask the authors (though I do not believe it is necessary for the main conclusion of this manuscript) if any of the data may suggest if the IL-7R is being degraded on its way to the plasma membrane (among other things, if a LIR is present it could be then anywhere in the protein) or if it is degraded once it is inserted in the membrane likely through degradation of endosomes containing IL7R (again, among other things, here we may expect a LIR, if present, to be in the extracellular region of the IL-7R that will become the cytosolic region once IL7R is in an endosome). It may be exciting to discuss those possibilities as if the degradation occurs on IL7R-containing endosomes, this may identify a new general mechanism through which selective autophagy using its known ability to degrade endosomes, may modulate signaling in T cells, which may extend to other signaling receptors that are also internalized in activated T cells.

Reviewer #3 (Remarks to the Author):

This is a thoughtful study that describes an elegant biochemical approach to identify autophagy targets in proliferating CD4 T cells. As autophagy is important for CD4 T cell proliferation and effector memory cell generation this work addresses an important topic and is very original. The work identifies IL-7R α as a target for degradation by autophagy and suggests that the accumulation of IL-7 receptor alpha chains would sequester the common gamma subunit of the IL-2 receptor complex and hence inhibit signalling via the IL-2 receptor. There is evidence in support of this hypothesis notably that CD4 T cells with defective autophagy show decreased ability to tyrosine phosphorylate STAT5 in response to IL2. However, there are some discrepancies in the experimental observations that are discordant with this model that need to be explored

1) IL-2 activation of STAT5 drives a positive feedforward loop that sustains expression of CD25.

However, the authors state that the expression of CD25 is normal in autophagy deficient CD4 T cells. They only show a single early time point for CD25 expression and possibly looking at longer time points to assess if autophagy deficient T cells can sustain CD25 expression is needed.

2) IL-2 is known to maintain T cell survival yet the authors state there are no survival defects in autophagy deficient CD4 T cells. How does this square with defects in IL2 signalling.

3) IL-2 controls expression of Myc and activation of mTORc1 and hence controls expression of glucose transporters, amino acid transporters and controls many aspects of T cell metabolism and cell growth. If autophagic deficient T cells have suppressed responses to IL-2 they will be small cells. This can be judged from their FACS scatter profiles. The authors should explore if these facets of IL-2 signalling are compromised by autophagy deficiencies,

Minor issues

i) In the in vitro proliferation experiments in Fig 1 cell counts should be shown to accurately disclose the impact of autophagy on T cell clonal expansion.

ii) The western blots showing the lipidation of LC3B are not of sufficient resolution. Are there better exposures that can better show the data?

iii) In the microscopy experiments how many cells were analysed? This is not clearly stated.

We are really grateful for the constructive comments from all three reviewers, which helped us to substantially improve our manuscript entitled 'Mapping of the autophagosomal degradome identifies IL-7R α as key cargo in proliferating CD4+ T cells' (# NCOMMS-21-46950A). We sincerely hope that our responses to the reviewers' comments will meet your requirements for publication. Thank you very much for your time and consideration. We are looking forward to your valuable feedback.

REVIEWER COMMENTS

Please find attached a reply to the reviewers (text in *italic*). We have highlighted new experiments and related text **in yellow** in the manuscript.

Reviewer #1:

Zhou *et al.* have developed an innovative mouse model introducing APEX2 into the LC3b locus, in order to employ proximity labelling and proteomics to detect autophagosomal substrates from primary cells *ex vivo*. The authors successfully employ this technique in activated CD4+ T lymphocytes and identify IL7R α as the most prominent target of the autophagy pathway in activated CD4+ T cells. In support, genetic deletion of Atg7 is found to increase cell surface expression of IL7R α on the cell surface of naïve but not effect CD4+ cells. The authors propose that this increase in IL7R α results in impaired T cell activation in Atg7-deficient cells by titrating the gamma chain away from IL2R complex. Overall, the studies are well done and support the conclusions that have been drawn. The APEX-LC3B mouse will be a useful resource to the autophagy field.

Some *additional mechanistic insight* into how IL7R α is being targeted for autophagic degradation in CD4 cells will greatly strengthen the manuscript. Since IL7R α is located at the cell surface or traffics through the endosomal recycling pathway, it is not entirely clear how such a protein would be captured into autophagosomes.

1) Although the identification of IL7R α via APEX2-LC3B proximity labelling implicates it as an autophagosome cargo, it is largely unclear how this receptor is being targeted for degradation via the autophagy pathway. Is the degradation of IL7R α dependent on a LIR-based interaction?

Yes, the murine IL-7R α contains a LIR domain at its intracellular domain (see iLIR <https://ilir.warwick.ac.uk/protein.php?organism=5&page=35>), from amino acid 399 to 404 (PVYQDL). Therefore, it is possible that the direct interaction between IL-7R α and LC3B mediates the autophagosomal degradation of IL-7R α .

2) Alternatively, are any of the known autophagy cargo receptors functionally required for IL7R α degradation? It is interesting that Figure 3 suggests that in contrast to MEFs, cargo receptors such as p62 and TAX1BP1 are not enriched in the autophagosomes of CD4+ T cells?

Based on the presence of a LIR domain in IL-7R α , it is likely that this molecule is recruited to autophagosomes through direct interaction with LC3B. Although not reaching statistical significance, p62/SQSTM1 was repeatedly found in BafA1 treated

pan-T cells activated for 3 days, example shown in **Fig. 1'** below. We reason that significance was not reached due to technical limitations, as mentioned in our current discussion below (page 19, line 453):

"In our study, activated primary CD4+ T cells show greater biological and technical variation than imMEFs. This is most likely due to the high amount of systemic variation in immune development and variation induced by activation/ expansion in T cells, which made this pilot study particularly challenging. In addition, T cells are smaller in size with little cytoplasm compared to imMEFs. As a result, a large number of T cells were required to map autophagy-regulated proteins."

Despite these limitations, it is striking that we consistently found IL-7R α as the most prominent target of autophagy in CD4+ T cells. Finally, we would like to thank the reviewer for this excellent question. We added the observation about a putative LIR domain being present in the intracellular domain of IL-7R α to our discussion (page 16, line 377) as follow:

"According to iLIR, a web resource that allows the prediction of Atg8-family interacting proteins, the murine IL-7R α contains a LIR domain in its intracellular domain from amino acid 399 to 404 (PVYQDL). Therefore, it is possible that IL-7R α is recruited to autophagosomes through direct interaction with LC3, either after its internalisation through the endosomal pathway or on its way to the plasma membrane."

Fig. 1' Volcano plot of proteins labelled by LC3B-AP2 in pan-T cells activated for 3 days. Proteins significantly upregulated in response to BafA1 treatment are highlighted in red, $p < 0.05$ by unpaired two-tailed Student's *t*-test, $n = 3$ biological replicates per group, with each replicate combining T cells from two mice.

3) In Figure 3e, why is there no difference in IL7R α protein levels between Atg7KO and control CD4 effector cells? Although the overall reduction in IL7R α compared to naïve CD4 cells makes sense, it is not clear why autophagy no longer involved in the turnover of IL7R in CD4 effector cells.

This is a very good point. According to a previous study, IL-7R α can be degraded by both lysosomal-dependent pathways and the Ubiquitin/proteasome system (UPS) (Henriques, Catarina M., et al. Blood, 2010). In effector T cells, the expression of IL-7R α is transcriptionally inhibited, thus stopping the production of new IL-7R α (Dooms, Hans, et al. JEM 2007). Moreover, during T cell activation, UPS is highly upregulated (Arata, Yoshiyuki, et al. Genes to Cells, 2019). Thus, we hypothesize that proteasome degradation might compensate for the lack of autophagy, resulting in degradation of any excessive IL-7R α in autophagy-deficient effector CD4+ T cells. As part of another study, we found a strong upregulation of many proteasomal subunits in autophagy-

deficient effector CD4⁺ T cells from gut by scRNAseq (unpublished). Aiming to address this hypothesis in this context, we treated activated autophagy-deficient T cells with the proteasome inhibitor MG-132. Indeed, IL-7R α level was significantly but mildly increased in activated autophagy-deficient CD4⁺ T cells when treated with 10 μ M MG-132 for 4 hours (Fig. 2'), indicating that IL-7R α can be degraded by the UPS pathway as a compensatory mechanism in activated Atg7-deficient T cells. Moreover, it also explains why autophagy-deficient effector CD4⁺ T cells showed no difference in IL-7R α level in comparison to their wild-type counterpart. We added this point to our discussion (page 16, line 364) as follows:

"We observed that IL-7R α levels were higher in autophagy-deficient naïve CD4⁺ T cells, while the difference disappears on day 3 post-activation. In effector T cells, the expression of IL-7R α is transcriptionally inhibited, thus stopping the production of new IL-7R α ¹. Moreover, previous studies indicated that IL-7R α can be degraded through both lysosome- and proteasome-mediated pathways in naïve T cells^{2,3}. It suggests that other compensatory pathways for degradation of excessive IL-7R α might take place. During T cell activation, genes of proteasome subunits are upregulated, which may contribute to the compensatory degradation of IL-7R α in autophagy-deficient CD4⁺ T cells⁴. Other pathways compensating gradually for the loss of autophagy in T-Atg7^{-/-} CD4⁺ T cells could be the explanation why the cells eventually catch up on their proliferation *in vitro*, and why IL-7R is not upregulated in the T-Atg7^{-/-} effector CD4⁺ T population."

Fig. 2' Levels of IL-7R α , gated on CD25⁺ CD4⁺ T cells from T-Atg7^{-/-} mice activated for 2 days, with or without MG-132 treatment for 4 hours. Each dot represents a biological replicate. Statistics by paired two-tailed Student's t-test, n = 5 mice per group, p* < 0.05.

4) In addition to ATG7, the authors observe similar phenotypes in ATG16KO cells. Both of these ATGs are part of the LC3 conjugation pathway, which mediates noncanonical LAP-like pathways such as LANDO, so it is not entirely clear from the results whether IL7R α is being targeted via classical autophagy. The authors should functionally evaluate IL7R α upon loss of an ATG involved in a distinct step of the classical autophagy trafficking process, such as ULK1/2 or ATG14.

Again, this is a very relevant point, and we would like to thank the reviewer for raising it. In Supplementary Figure 3h, we showed that the internalisation rate of IL-7R α was not changed in Atg7-deficient naïve T cells, implying that the accumulation of IL-7R α is not due to defects in LAP or LANDO. However, to further confirm this point and in the absence of mice specifically lacking the expression of these molecules in T cells in our lab, we treated wild-type CD4⁺ T cells with autophagy inhibitors *in vitro*, which typically block autophagy at the initiation phase (e.g. SBI-0206965, an ULK1/2 inhibitor) (Alsaleh, Ghada, *et al.* *Elife*, 2020) or at the nucleation phase (e.g. 3-methyladenine,

a PI3K inhibitor) (Mocholi, Enric, *et al.* Cell reports, 2018), and measured IL-7R α expression. As **Fig. 3'** shows, IL-7R α levels were significantly increased in naïve CD4 $^{+}$ T cells treated with either 2.5 μ M SBI-0206965 (ULK inhibitor), or 10 μ M 3-methyladenine (3-MA), for 20 hours, further supporting our finding that IL-7R α is degraded through classical autophagy pathway. These observations have been added in **Supplementary Fig. 3i** and **j**, and the text describing it in our manuscript (page 13, line 287) as below:

“Moreover, surface and intracellular protein levels of IL-7R α were significantly increased in wild-type naïve CD4 $^{+}$ T cells when treated with SBI-0206965 (Supplementary Fig. 3i), an ULK1/2 inhibitor blocking the initiation phase of autophagy⁵, and 3-methyladenine (3-MA) (Supplementary Fig. 3j), a PI3K inhibitor blocking the nucleation phase of autophagy⁶. This further confirms that IL-7R α is degraded through classical autophagy.”

Fig. 3' Levels of IL-7R α gated on naïve CD4 $^{+}$ T cells from wild-type mice, with or without 2.5 μ M SBI-0206965 (**a**, ULK inh) or 10 μ M 3-methyladenine (**b**, 3-MA) treatment for 20 hours. Each dot represents a biological replicate. Statistics by paired two-tailed Student's *t*-test, $n = 4$ mice per group, $p^* < 0.05$.

Reviewer #2:

This is a very interesting study where the authors not only present evidence of the generation of a new mouse model that can become a very powerful tool to identify cargo degraded by autophagy in different cell types under different situations using a much easier and in many occasions better method than classical autophagosome isolation, but it also uses that tool to identify IL7R degradation as an important process that may explain how autophagy is able to regulate T cell proliferation and differentiation. The model has been carefully evaluated and optimized to permit the identification of autophagosome cargo minimizing background due to interaction that may occur outside of the autophagosome where LC3 can also be found. The data that supports that IL-7 receptor is degraded by autophagy in activated T cells and may interfere with IL-2 signaling is also conclusive and, as mentioned above, the analysis of all the data obtained from the proteomic analysis may even open the door for further identification of autophagosome cargo in T cells and expand our current knowledge of how autophagy regulates T cells. I just would like to point a few minor issues that I believe, if addressed, may help strengthen some of the conclusions and get a better glimpse of the implications of this study for T cell biology.

1. When the authors show the experiments that have been performed as part of their goal to validate the new mouse model, it could be important that they would offer a possible explanation as to why only partial protection and not almost total preservation of p62 (easier to understand) and AP2 (maybe needs some explanation) occurs in cells treated only with PK vs PK + Tx.

This is a very useful comment, prompting us to improve the description of this figure in the text. Although LC3B-AP2 and p62 are mainly enriched inside the autophagosome, they are also localised on the cytoplasmic side of autophagosomes. This explains why we see the partial preservation of these proteins by PK, since only the ones residing in autophagosomes are protected by their double-membrane structure whereas PK + Tx degrades these proteins everywhere. A similar result was also observed by Le Guerroué, François, et al. Molecular Cell 2017 (Fig. 4'a) and Zellner, Susanne., et al. Molecular Cell 2021 (Fig. 4'b). Based on their observations and to limit the proximity labelling by LC3B-AP2 to proteins inside the autophagosomes, we used proteinase K before enriching biotin-labelled proteins and proceeding to mass spectrometry analysis (Supplementary Fig. 2d). We added this point to our manuscript (page 9, line 185) as follow:

“p62, an autophagy cargo receptor protein, and LC3-AP2 (and their proximity-biotinylated proteins) are located both inside and outside of autophagosomes. Prot K can only digest the proteins outside the autophagosomes, having no access to the inside ones which are protected by the intact double-membrane structure. Therefore, and similar to previous observations in LC3B-AP2 overexpression systems in vitro^{7,8}, the sample treated with only Prot K partially preserved p62, LC3-AP2 and biotinylated bands compared to untreated or Triton-X treated control, while the cytosolic tail of the transmembrane protein ATG9A was completely degraded (Fig. 2e).”

Fig. 4' Original Figures from Le Guerroué, François, *et al.* *Molecular Cell* 2017 (a) and Zellner, Susanne., *et al.* *Molecular Cell* 2021 (b)

(a) Homogenates from APEX2-hATG8s chimera expressing HeLa cells grown in the presence of BafA1 (2 hr), BP (30 min), and H₂O₂ (1 min) were left untreated or incubated with proteinase K, Triton X-100, or both followed by immunoblotting.

(b) APEX2 bait-expressing HeLa cells were treated for 2 h with BafA1, 30 min with biotin-phenol, and 1 min with H₂O₂ followed by homogenization. Homogenates were incubated with proteinase K, Triton X-100, or both and immunoblotted for APEX, biotin, p62, and ATG9. The latter 2 served as controls.

2. Also in this regard, a quantification similar to the one they offer in some of their other colocalization studies, could be offered for Fig. 2, just to get a more accurate idea of how many of the biotinylated proteins do actually colocalize with AP2 in the autophagosome or LAMP1 in the autolysosome.

Following the suggestion of Reviewer #2, we analysed the Pearson's correlation coefficient of biotin with both AP2 and LAMP1, as shown below (this has now been added in the text, figures, and figure legends).

Biotin largely colocalises with **AP2** (Pearson's coefficient over **0.8**) and, as expected, this was not impacted by BafA1 administration (Fig. 5'a). These results are consistent with a previous study indicating that radicals generated by AP2 are short-lived, and only biotinylate proximal endogenous proteins (Lam, Stephanie S., *et al.* *Nature methods* 2015) and therefore, the biotin and AP2 signals largely overlap.

Since not all autophagosomes are fused with lysosomes and form autolysosomes at once, we only observed a partial co-localisation of **biotin** with **LAMP1** (Fig. 5'b, Pearson's coefficient around **0.3**). Moreover, when imMEFs were treated with BafA1, a drug blocking the fusion between autophagosomes and lysosomes⁹, the co-localisation between these two molecules was significantly decreased as expected. These graphs are included in the **Supplementary Fig. 2c** and addressed in manuscript (page 8, line 169) as below:

"In the imMEFs, as expected, we observed the co-localisation of biotinylated proteins with AP2 and LAMP1, a lysosomal marker (Fig. 2c), further confirming the autophagosomal enrichment of LC3B-AP2. Biotin largely colocalises with AP2 (Pearson's coefficient over 0.8) (Supplementary Fig. 2c) which indicates proteins labelled with biotin are proximal to AP2 enzymes. In contrast, since not all autophagosomes are fused with lysosomes at once, biotin was only partially co-localised with LAMP1 (Supplementary Fig. 2c, Pearson's coefficient around 0.3). As expected, when imMEFs were treated with BafA1, the co-localisation between these two molecules was significantly decreased."

Fig. 5' Bar graphs show Pearson's correlation coefficient between Biotin and AP2 (a), as well as Biotin and LAMP1 (b), $n = 19-26$ cells per group. Data are represented as mean \pm SEM, with unpaired two-tailed Student's t -test, $p^* < 0.05$, ns: not significant.

3. The BM chimeras experiment offer new data that support a central role of autophagy in the CD4+ responses against infection. It is a pity, though, that using MHC-II tetramers in those experiments we cannot know how those effects on CD4+ T cells may also extend to the antiviral CD8+ T cell response in vivo. **Did the authors collected any data in their FACS analyses that may shed some light into that?** This may be important based on previous reported data that appears to suggest that acute expansion on **CD8+ T cells following viral infection may not be modulated by autophagy in vivo**, suggesting that autophagy may be more important to CD4+ than CD8+ T cell expansion in vivo.

We did not collect any data on the effects of autophagy in CD8+ T cells because this had been shown before in similar BM chimera set-ups by us (Puleston, Daniel J., et al. Elife 2014, Fig. 3) and others (Xu, Xiaojin, et al. Nature immunology 2014, and Schlie, Katrin, et al. The Journal of Immunology 2015). Unlike CD4+ T cells, autophagy-deficient CD8+ T cells do not display an impaired effector expansion during the primary infection. Instead, in CD8+ T cells, the defects appeared only later during the effector-to-memory cell transition phase. That is why we chose to explore autophagy's role in the effector formation of CD4+ rather than CD8+ T cells here. On the other hand, this study was foremost a proof-of-principle to test whether the proximity labelling mouse model can be used for the identification of autophagosomal content in primary cells. We stressed in the manuscript that indeed autophagy is more important to CD4+ than CD8+ T cell expansion in vivo in our discussion part (page 17, line 403) as follow:

“Additionally, autophagy has been shown to be more important in the effector expansion of CD4+ rather than CD8+ T cells, while in CD8+ T cells, the defects appeared only later during the effector-to-memory cell transition phase, which also negatively impacts the secondary anti-viral response to the same pathogen¹⁰⁻¹².”

4. There are a couple of implications that I think may be quite exciting to expand upon and the authors may consider including them in their discussion. First, it would be the implications that this mechanism may have on **T cell differentiation**, especially in two specific aspects. First, in line with data the authors had already published, do they think that persistence of IL7R may result, as they show here, in reduced IL2R and IL15R signaling that may account, at least in part, for the defects in **the efficient generation of memory they recently reported in a very elegant study**. Second, there may be **an effect of autophagy on CD4+ T cell effector differentiation** based on how it can regulate IL7R levels. One can envision, for example, a situation where loss of autophagy resulting in reduced common gamma chain receptor would inhibit processes such as Th2 differentiation that rely on signaling from two common gamma chain receptors, IL2R and IL4R; while in other processes such as Th17 differentiation, we may

see an opposite effect. As the negative effect of IL2R signaling would be reduced, while there would be no effect on signaling from other cytokine receptors (not members of the common gamma chain family) that direct Th17 generation.

*These are very relevant hypotheses, and we absolutely would like to integrate them into our discussion. Autophagy has been long considered to control the differentiation of T cells through degrading key transcription factors (e.g. PU.1 for Th9 cells) or modulating metabolic pathways (memory T cells, Tregs and so on). It is possible that its impact on IL-2/IL-15 signalling during early activation phase also influences the fate decision of T cells. For memory CD8+ T cells, it has been shown that IL-15, whose receptor is composed of IL-15 α , IL-2 β and γ_c , is crucial for the homeostatic maintenance of the memory pool (Nolz, Jeffrey C., and Martin J. Richer. *Molecular Immunology* 2020). As Gros *et al.* showed in their recent study, the formation of memory CD4+ T cells is impaired without autophagy (Murera, Diane, *et al.* *Scientific reports* 2018). An IL-2 signalling-dependent mechanism might also contribute to this phenotype, since IL-2 has been shown to be essential for CD4+ T-cell memory formation (McKinstry, K. Kai, *et al.* *Nature communications* 2014). Moreover, IL-2 signalling is antagonistic to Th17 differentiation through suppressing the expression of IL-6R β (or gp130). Consistent with a previous study which shows an expansion of IL-17-producing cells in Lck^{Cre}, Atg7^{fllox/fllox} mice (Amersfoort, Jacob, *et al.* *Frontiers in immunology* 2018), the decreased IL-2 signalling in autophagy-deficient CD4+ T cells may facilitate its differentiation into Th17.*

*By contrast, although the differentiation of Th2 cells requires both IL-2 and IL-4, two common gamma chain cytokines, in fact, a study we contributed to shows that autophagy-deficiency has no effect on Th2 cells (Kabat, A. M., Riffelmacher, T., ... Simon, A. K. & Maloy, K. J. *Elife* 2016). Since autophagy-deficient CD4+ T cells have a lower level of IL-2 signalling rather than a total lack of it, it is possible that Th2 differentiation does not require strong IL-2 signalling as required by Th0 expansion. Further investigation into how the strength of common gamma chain signalling affect CD4+ T cell fate decision may provide an answer. This interesting point is added to the discussion (page 17, line 394) as follow.*

*“Moreover, autophagy may contribute to the differentiation and memory formation of CD4+ T cells through modifying common gamma chain cytokine signalling (IL-2/IL-15). IL-2 signalling is antagonistic to Th17 differentiation since it suppresses the expression of IL-6R β (or gp130)^{13,14}. Consistent with a previous study which shows an expansion of IL-17-producing cells in Lck^{Cre}, Atg7^{fllox/fllox} mice¹⁵, the decreased IL-2 signalling in autophagy-deficient CD4+ T cells may facilitate its differentiation into Th17. Also, as Gros *et al.* showed in their recent study, the formation of memory CD4+ T cells is impaired without autophagy¹⁶. An IL-2 or IL-15 signalling-dependent mechanism might also contribute to this phenotype, since IL-2 has been shown to be essential for CD4+ T-cell memory formation and IL-15 for memory maintenance^{17,18}.”*

5. Finally, it may be interesting if the authors could **analyze the IL7Ra sequence** to see if it has putative LIRs, that would support its interaction with LC3, and where they are. Though their absence would not preclude the conclusions reached here at all (it could all be autophagy receptor mediated) if present, it may help understand how this process occurs. In that sense, I wanted to ask the authors (*though I do not believe it is necessary for the main conclusion of this manuscript*) if any of the data may suggest if the IL-7R is **being degraded on its way to the plasma membrane** (among other things, if a LIR is present it could be then anywhere in the protein) or if it is **degraded once it is inserted in the membrane** likely through degradation of endosomes containing IL7R (again, among other things, here we may expect a LIR, if present, to be in the **extracellular region** of the IL-7R that will become the cytosolic

region once IL7R is in an endosome). It may be exciting to discuss those possibilities as if the degradation occurs on IL7R-containing endosomes, this may identify a new general mechanism through which **selective autophagy using its known ability to degrade endosomes**, may modulate signaling in T cells, which may extend to other signaling receptors that are also internalized in activated T cells.

Yes, according to iLIR, a web resource that allows the prediction of Atg8-family interacting proteins, the murine IL-7R α contains a LIR domain from amino acid 399 to 404 (PVYQDL) (iLIR: <https://ilir.warwick.ac.uk/protein.php?organism=5&page=35>). Surprisingly, this domain is intracellular. When activating CD4⁺ T cells for 3 days, we observed the colocalization of Rab7/CD63 (late endosome markers), LC3, and IL-7R α (Fig. 6' and 7'), implying that the autophagy-mediated degradation of IL-7R α may involve autophagy of late endosomes. However, the process is less likely to involve the invagination of IL-7R α into the late endosomal lumen and the formation of amphisomes (fusion of autophagosome with endosome), since the interaction of IL-7R α with LC3B in this context requires a LIR domain at its extracellular region, as the reviewer suggested. Based on the data described and presented below, we hypothesized a model of how IL-7R α might be degraded by autophagy (Fig. 8'). Besides, IL-7R α may also be degraded on its way to the plasma membrane to limit the surface IL-7R α levels in effector T cells. As the reviewer suggested, the location of the LIR domain is flexible in this context. We include this point in our discussion (page 16, line 377) as follow:

“According to iLIR, a web resource that allows the prediction of Atg8-family interacting proteins, the murine IL-7R α contains a LIR domain in its intracellular domain from amino acid 399 to 404 (PVYQDL). Therefore, it is possible that IL-7R α is recruited to autophagosomes through direct interaction with LC3, either after its internalisation through the endosomal pathway or on its way to the plasma membrane.”

Day 3 activation, CD4⁺ T cells

Fig. 6' Confocal imaging of IL-7R α (CD127), LC3 and CD63 (a late endosome marker) in wild-type CD4⁺ T cells activated for 3 days. Red arrows indicate colocalization events, whose magnifications are displayed at the upper-right corners.

Day 3 activation, CD4+ T cells

Fig. 7' Confocal imaging of IL-7R α (CD127), LC3 and Rab7 (a late endosome marker) in wild-type CD4+ T cells activated for 3 days. Red arrows indicate colocalization events, whose magnifications are displayed at the upper-left corners.

Fig. 8' Proposed mechanism of how IL-7R α directly binds to LC3 and is degraded through autophagy. IL-7R α is internalised and enters the early and late endosome sequentially. The cytosolic LIR domain interacts directly with LC3 molecules residing on the phagophore double-membrane structure, which facilitates its autophagy-mediated degradation.

Reviewer #3:

This is a thoughtful study that describes an elegant biochemical approach to identify autophagy targets in proliferating CD4 T cells. As autophagy is important for CD4 T cell proliferation and *effector memory cell generation*, this work addresses an important topic and is very original. The work identifies IL-7R α as a target for degradation by autophagy and suggests that the accumulation of IL-7 receptor alpha chains would sequester the common gamma subunit of the IL-2 receptor complex and hence inhibit signalling via the IL-2 receptor. There is evidence in support of this hypothesis notably that CD4 T cells with defective autophagy show decreased ability to tyrosine phosphorylate STAT5 in response to IL2. However, there are some discrepancies in the experimental observations that are discordant with this model that need to be explored:

1) IL-2 activation of STAT5 drives a positive feedforward loop that sustains expression of CD25. However, the authors state that the expression of CD25 is normal in autophagy deficient CD4 T cells. They only show a single early time point for CD25 expression and possibly looking at longer time points to assess if autophagy deficient T cells can sustain CD25 expression is needed.

*Although we didn't find an impaired upregulation of CD25 expression at an early time point (day 1) of T cell activation, this phenotype could appear later during activation as suggested by the reviewer. We activated CD4+ T cells in vitro and observed that CD25 expression was indeed significantly decreased in autophagy-deficient cells at days 3 post-activation (Fig. 9'). We hypothesize that early CD25 expression (day 1) is primarily controlled by TCR/CD28 signalling, while IL-2 signalling takes it over at later stage (day 3) as the high affinity IL-2 receptors get upregulated. Therefore, attenuated IL-2 signalling in T-Atg7 CD4+ T cells only affects its CD25 expression on day 3 activation. We have addressed that in **Supplementary Fig. 4b** and in the manuscript (page 14, line 327) as follow:*

"However, due to the impaired IL-2 signalling, the expression levels of IL-2R α , controlled by IL-2 signalling, were significantly decreased in T-Atg7^{-/-} T cells on day 3 post activation (Supplementary Fig. 4b)."

Fig. 9' Bar graph of CD25 surface level, gated on activated CD4+ T cells activated for 1 or 3 days in vitro. All values represented as mean \pm SEM with two-tailed unpaired Student's *t*-test, *n* = 4 mice per group, **p* < 0.05, ns: not significant.

2) IL-2 is known to maintain T cell survival yet the authors state there are no survival defects in autophagy deficient CD4 T cells. How does this square with defects in IL-2 signalling.

It has been previously shown by two ground-breaking studies that IL-2 signalling is crucial for both T cell proliferation (Doreen A., and Kendall A. Smith. Science 1984)

and cell survival during activation (Vella, Anthony T., *et al.* PNAS 1998). In our original data, we noticed a trend, though not significant (probably due to the large variation), in the increase of cell death in autophagy-deficient CD4⁺ T cells post activation. Therefore, we combined the results from two independent experiments. Indeed, as **Fig. 10'** shown below, autophagy-deficient CD4⁺ T cells show a significant but mild (around 10 %) increase in their cell death on day 3 post activation. It further corroborates our observation that autophagy-deficient CD4⁺ T cells displayed an attenuated IL-2 signalling. The reason why proliferation is more affected than survival is probably that the normal proliferation of CD4⁺ T cells requires higher levels of IL-2 signalling than survival. Autophagy-deficient CD4⁺ T cells, rather than totally lacking IL-2 signalling such as CD25- or Il-2 knockout models, exhibit only diminished IL-2 signalling, which has a bigger impact on cell proliferation than survival. We are really grateful that reviewer 3 kindly pointed this out, and accordingly, we made a change in **Supplementary Fig. 1i** and manuscript as follow (page 6, line 116):

“Moreover, decreased expansion cannot be solely explained by increased cell death, since the percentages of CD4⁺ T cells did not differ significantly between wild-type and knockout cells on day 1 post activation, and it only displayed a significant but mild increase on day 3 (Supplementary Fig. 1i). Since the survival of activated T cells is modulated by multiple signalling pathways^{19,20}, this also implies certain deficits in these pathways when cells are lacking autophagy.”

Fig. 10' Histograms (left) shown LD-NIR⁺ population within CD4⁺ T cells, on day 1 and 3 of activation *in vitro*. Bar graph (right) indicates the percentage of gated population within live CD4⁺ population ($n = 8$ mice per group).

3) IL-2 controls expression of Myc and activation of mTORC1 and hence controls expression of glucose transporters, amino acid transporters and controls many aspects of T cell metabolism and cell growth. If autophagic deficient T cells have suppressed responses to IL-2, they will be small cells. This can be judged from their FACS scatter profiles. The authors should explore if these facets of IL-2 signalling are compromised by autophagy deficiencies.

We agree that autophagy-deficient CD4⁺ T cells may display metabolic defects due to impaired IL-2 signalling (Preston, Gavin C., *et al.* The EMBO journal, 2015)(Marchingo, Julia M., *et al.* Elife, 2020). Following the reviewer’s suggestion, we further explored the expression of phospho-S6 (downstream of mTORC1), c-MYC, GLUT-1 (a glucose transporter), CD98 (an amino acid transporter), and cell size of CD4⁺ T cells post-activation for 1 day through flow cytometry. Consistent with the phenotype of attenuated IL-2 signalling, the levels of phospho-S6, c-MYC and CD98 were significantly decreased in autophagy-deficient CD4⁺ T cells, reflecting compromised upregulation of their cell metabolism upon activation (**Fig. 11'**). Interestingly, we notice that both cell size and GLUT-1 expression levels were significantly increased. It has

been shown that inducible deletion of autophagy-related genes prevents the reduction of cell size in cell lines upon starvation (which is a treatment to boost autophagy), probably due to the inhibition of recycling cellular content (Hosokawa, N., Hara, Y., & Mizushima, N. *FEBS letters*, 2006). This effect may contribute to the cell size change in autophagy-deficient CD4⁺ T cells during activation. Also, our lab previously reported that autophagy-deficient CD8⁺ T cells (Puleston, D. J., ... & Simon, A. K., *Elife*, 2014) and leukemic cells (Watson, A. S., Riffelmacher, T., ... & Simon, A. K., *Cell death discovery*, 2015) display increased surface GLUT1 expression. It may reflect a disrupted glucose metabolism of T cells adapted to the status of autophagy deficiency. These results indicate that T-Atg7^{-/-} CD4⁺ T cells may have undergone some long-term adaptive changes in the overall network of signalling pathways which in turn affect different physiological aspects. We include the results of c-MYC and mTORC1 (phospho-S6), since they are two direct downstream targets of IL-2 signalling, as the reviewer suggested, which may help reflect the direct consequences of changes in IL-2 signalling. This further corroborates our observation of impaired IL-2 signalling in T-Atg7^{-/-} CD4⁺ T cells. They are shown by Supplementary Fig. 4d and e, and in our manuscript as follow (page 15, line 344):

“Moreover, c-MYC and mTORC1, two direct downstream targets of IL-2 signalling pathway which govern T cell metabolism^{21,22}, displayed a significant decrease in autophagy-deficient CD4⁺ T cells during activation (Supplementary Fig. 4d and e). This may also contribute to their delayed proliferation upon antigen stimulation.”

Fig. 11' Metabolic pathways are changed in T-Atg7^{-/-} CD4⁺ T cells.

(a) Bar graph shows the fold change in phospho-S6 geometric mean fluorescence intensity of CD25+ CD4+ T cells (activated for 1 day) from T-Atg7^{-/-} mice normalised to the ones from wild-type mice, n = 8 mice per group.

(b) Bar graph shows the c-MYC geometric mean fluorescence intensity of CD25+ CD4+ T cells activated for 1 day, from both wild-type and T-Atg7^{-/-} mice, n = 4 mice per group.

(c) Bar graph shows the fold change in surface CD98 geometric mean fluorescence intensity of CD25+ CD4+ T cells (activated for 1 day) from T-Atg7^{-/-} mice normalised to the ones from wild-type mice, n = 8 mice per group.

(d) Bar graph shows the fold change in forward scatter area (FSC-A) geometric mean fluorescence intensity of CD25+ CD4+ T cells (activated for 1 day) from T-Atg7^{-/-} mice normalised to the ones from wild-type mice, n = 8 mice per group.

(e) Bar graph shows the surface GLUT-1 geometric mean fluorescence intensity of CD25+ CD4+ T cells from both wild-type and T-Atg7^{-/-} mice, n = 4 mice per group. Cells were stained with the AF488 conjugated HTLV receptor binding domain (H_{RBD}), that binds GLUT-1.

All values were represented as mean ± SEM with unpaired two-tailed Student's t-test, p* < 0.05, p** < 0.01, p*** < 0.0001.

Minor issues

i) In the in vitro proliferation experiments in Fig 1, cell counts should be shown to accurately disclose the impact of autophagy on T cell clonal expansion. (actually having more cells or not).

We agree this is necessary data to determine the impact of autophagy on T cell expansion. We repeated the in vitro proliferation experiment and added equal number of cells into each well for activation and calculate the absolute count of CD4+ T cells. The result is shown as Fig. 12' below. We also included it in our manuscript as Supplementary Fig. 1e. We notice that while autophagy-deficient CD4+ T cells (T-Atg7^{-/-}) starts to proliferate between day 3 and 7, their total cell number is still about 25% lower than its wildtype counterpart, which is consistent with the observation of their late onset proliferation.

Fig. 12' Splenic OT-II+ T cells from wild-type or T-Atg7^{-/-} mice were stimulated with ovalbumin in culture medium supplemented with mIL2 for 1, 3, 7 or 10 days. Histograms by flow cytometry (left) represent OT-II+ CD4+ T cell proliferation. Number indicates the total cell number of gated population in million(s).

ii) The western blots showing the lipidation of LC3B are not of sufficient resolution. Are there better exposures that can better show the data?

We replaced the original **Fig. 2b** with **Fig. 13'** to increase the resolution of image, and made this change in the manuscript (page 7, line 150):

“Immunoblotting of splenocytes showed bands around 44 kDa (LC3B: 17 kDa, AP2: 27 kDa) in heterozygotes and homozygotes with anti-LC3B staining, but not in wild-type cells (Fig. 2b).”

Fig. 13' Western blot of whole splenic cell homogenates from wild-type, heterozygous and homozygous mice. Cells were treated with 10 nM Bafilomycin A1 (BafA1) or DMSO for 2 hours.

iii) In the microscopy experiments how many cells were analysed? This is not clearly stated.

We want to thank the reviewer for this very crucial point. We added the statistics analysis and number of cells being analysed as follow, also in **Supplementary Fig. 2c, Fig. 3e and f** in our manuscript.

Statistics for Fig. 2c, which is added as **Supplementary Fig. 2c** and in our manuscript (page 8, line 169) as below:

“In the imMEFs, as expected, we observed the co-localisation of biotinylated proteins with AP2 and LAMP1, a lysosomal marker (Fig. 2c), further confirming the autophagosomal enrichment of LC3B-AP2. Biotin largely colocalises with AP2 (Pearson’s coefficient over 0.8) (Supplementary Fig. 2c) which indicates proteins labelled with biotin are proximal to AP2 enzymes. In contrast, since not all autophagosomes are fused with lysosomes at once, biotin was only partially co-localised with LAMP1 (Supplementary Fig. 2c, Pearson’s coefficient around 0.3). As expected, when imMEFs were treated with BafA1, the co-localisation between these two molecules was significantly decreased.”

Fig. 5' (same as above). Bar graphs show Pearson's correlation coefficient between Biotin and AP2 (a), as well as Biotin and LAMP1 (b), **n = 19-26 cells per group**. Data are represented as mean \pm SEM, with unpaired two-tailed Student's t-test, $p^* < 0.05$, ns: not significant.

Statistics for Fig. 3d, which is added as **Fig. 3e and f**, and in our manuscript (page 12, line 261) as below:

"We observed colocalization of IL-7R α with both molecules when cells are either activated or not (Fig. 3d), and the colocalization is increased upon BafA1 treatment (Fig. 3e and f)."

Fig. 14' Bar graphs show Mander's correlation coefficient between IL-7R α and WIPI2, as well as IL-7R α and LC3, on non-activated wild-type CD4+ T cells or activated for 3 days, **n = 24-40 cells per group**. Data are represented as mean \pm SEM, with unpaired two-tailed Student's t-test, $p^{**} < 0.01$, $p^{***} < 0.001$, $p^{****} < 0.0001$ ns: not significant.

As for **Fig. 4d** and **e**, cell numbers were specified in the figure legends in our original manuscript, with **$n = 13-20$ cells per group** and **$n = 23$ cells per group** respectively.

References

- 1 Dooms, H., Wolslegel, K., Lin, P. & Abbas, A. K. Interleukin-2 enhances CD4⁺ T cell memory by promoting the generation of IL-7R alpha-expressing cells. *J Exp Med* **204**, 547-557, doi:10.1084/jem.20062381 (2007).
- 2 Henriques, C. M., Rino, J., Nibbs, R. J., Graham, G. J. & Barata, J. T. IL-7 induces rapid clathrin-mediated internalization and JAK3-dependent degradation of IL-7Ralpha in T cells. *Blood* **115**, 3269-3277, doi:10.1182/blood-2009-10-246876 (2010).
- 3 Faller, E. M., Ghazawi, F. M., Cavar, M. & MacPherson, P. A. IL-7 induces clathrin-mediated endocytosis of CD127 and subsequent degradation by the proteasome in primary human CD8 T cells. *Immunol Cell Biol* **94**, 196-207, doi:10.1038/icb.2015.80 (2016).
- 4 Arata, Y. *et al.* Defective induction of the proteasome associated with T-cell receptor signaling underlies T-cell senescence. *Genes Cells* **24**, 801-813, doi:10.1111/gtc.12728 (2019).
- 5 Alsaleh, G. *et al.* Autophagy in T cells from aged donors is maintained by spermidine and correlates with function and vaccine responses. *Elife* **9**, doi:10.7554/eLife.57950 (2020).
- 6 Mocholi, E. *et al.* Autophagy Is a Tolerance-Avoidance Mechanism that Modulates TCR-Mediated Signaling and Cell Metabolism to Prevent Induction of T Cell Anergy. *Cell Rep* **24**, 1136-1150, doi:10.1016/j.celrep.2018.06.065 (2018).
- 7 Le Guerroue, F. *et al.* Autophagosomal Content Profiling Reveals an LC3C-Dependent Piecemeal Mitophagy Pathway. *Mol Cell* **68**, 786-796 e786, doi:10.1016/j.molcel.2017.10.029 (2017).
- 8 Zellner, S., Schifferer, M. & Behrends, C. Systematically defining selective autophagy receptor-specific cargo using autophagosome content profiling. *Mol Cell* **81**, 1337-1354 e1338, doi:10.1016/j.molcel.2021.01.009 (2021).
- 9 Mauvezin, C., Nagy, P., Juhasz, G. & Neufeld, T. P. Autophagosome-lysosome fusion is independent of V-ATPase-mediated acidification. *Nat Commun* **6**, 7007, doi:10.1038/ncomms8007 (2015).
- 10 Xu, X. *et al.* Autophagy is essential for effector CD8(+) T cell survival and memory formation. *Nat Immunol* **15**, 1152-1161, doi:10.1038/ni.3025 (2014).
- 11 Puleston, D. J. *et al.* Autophagy is a critical regulator of memory CD8(+) T cell formation. *Elife* **3**, doi:10.7554/eLife.03706 (2014).
- 12 Schlie, K. *et al.* Survival of effector CD8⁺ T cells during influenza infection is dependent on autophagy. *J Immunol* **194**, 4277-4286, doi:10.4049/jimmunol.1402571 (2015).
- 13 Laurence, A. *et al.* Interleukin-2 signaling via STAT5 constrains T helper 17 cell generation. *Immunity* **26**, 371-381, doi:10.1016/j.immuni.2007.02.009 (2007).
- 14 Yang, X. P. *et al.* Opposing regulation of the locus encoding IL-17 through direct, reciprocal actions of STAT3 and STAT5. *Nat Immunol* **12**, 247-254, doi:10.1038/ni.1995 (2011).
- 15 Amersfoort, J. *et al.* Defective Autophagy in T Cells Impairs the Development of Diet-Induced Hepatic Steatosis and Atherosclerosis. *Front Immunol* **9**, 2937, doi:10.3389/fimmu.2018.02937 (2018).
- 16 Murera, D. *et al.* CD4 T cell autophagy is integral to memory maintenance. *Sci Rep* **8**, 5951, doi:10.1038/s41598-018-23993-0 (2018).
- 17 McKinsty, K. K. *et al.* Effector CD4 T-cell transition to memory requires late cognate interactions that induce autocrine IL-2. *Nat Commun* **5**, 5377, doi:10.1038/ncomms6377 (2014).

- 18 Purton, J. F. *et al.* Antiviral CD4⁺ memory T cells are IL-15 dependent. *J Exp Med* **204**, 951-961, doi:10.1084/jem.20061805 (2007).
- 19 Vella, A. T., Dow, S., Potter, T. A., Kappler, J. & Marrack, P. Cytokine-induced survival of activated T cells in vitro and in vivo. *Proc Natl Acad Sci U S A* **95**, 3810-3815, doi:10.1073/pnas.95.7.3810 (1998).
- 20 Zhan, Y., Carrington, E. M., Zhang, Y., Heinzl, S. & Lew, A. M. Life and Death of Activated T Cells: How Are They Different from Naive T Cells? *Front Immunol* **8**, 1809, doi:10.3389/fimmu.2017.01809 (2017).
- 21 Preston, G. C. *et al.* Single cell tuning of Myc expression by antigen receptor signal strength and interleukin-2 in T lymphocytes. *EMBO J* **34**, 2008-2024, doi:10.15252/embj.201490252 (2015).
- 22 Marchingo, J. M., Sinclair, L. V., Howden, A. J. & Cantrell, D. A. Quantitative analysis of how Myc controls T cell proteomes and metabolic pathways during T cell activation. *Elife* **9**, doi:10.7554/eLife.53725 (2020).

REVIEWERS' COMMENTS

Reviewer #1 (Remarks to the Author):

I have reviewed the revised manuscript and the authors' response. My previous concerns have been satisfied and I think the manuscript is an important advance for the field. The LIR in the IL17a cytoplasmic domain is interesting; it should be validated experimentally but this can best be pursued in a future study.

Reviewer #2 (Remarks to the Author):

This revised version of the manuscript by Zhou et al has addressed all the small concerns this reviewer had with the original version, and it now not only describes a new mouse model to identify autophagy substrates in T cells and in many other cell types; but it also identifies the IL-7 receptor as an important autophagy substrate, whose regulation by autophagy can account for the decreased proliferative ability that has been described by many groups on CD4+ T cells that lack essential autophagy genes.

The authors have included new analysis to better understand several concepts (e.g., magnitude of the targeting of certain substrates to autophagosomes), identify LIR motifs in the IL-7 receptor, offering possible mechanisms through which this protein may be degraded by autophagy); and add a discussion of possible implications for different aspects of T cell function, of the regulation of IL-7 by autophagy.

This work, therefore, provides an extremely useful tool to better study autophagy in T (and other) cells and identifies IL-7 receptor as a key substrate of autophagy in T cells that may explain not only how autophagy control T cell proliferation but also other aspects of T cell biology that have been shown to be regulated by autophagy.

Reviewer #3 (Remarks to the Author):

I thank the authors for their careful responses to my comments

We are really grateful for the constructive comments from all three reviewers previously, which helped us to substantially improve our manuscript entitled 'Mapping autophagosome contents identifies interleukin-7 receptor- α as a key cargo modulating CD4+ T cell proliferation' (# NCOMMS-21-46950B). Here, we include further comments from all three reviewers.

REVIEWERS' COMMENTS

Reviewer #1 (Remarks to the Author):

I have reviewed the revised manuscript and the authors' response. My previous concerns have been satisfied and I think the manuscript is an important advance for the field. The LIR in the IL17a cytoplasmic domain is interesting; it should be validated experimentally but this can best be pursued in a future study.

We agree with this reviewer that this should be validated in a further study.

Reviewer #2 (Remarks to the Author):

This revised version of the manuscript by Zhou et al has addressed all the small concerns this reviewer had with the original version, and it now not only describes a new mouse model to identify autophagy substrates in T cells and in many other cell types; but it also identifies the IL-7 receptor as an important autophagy substrate, whose regulation by autophagy can account for the decreased proliferative ability that has been described by many groups on CD4+ T cells that lack essential autophagy genes.

The authors have included new analysis to better understand several concepts (e.g., magnitude of the targeting of certain substrates to autophagosomes), identify LIR motifs in the IL-7 receptor, offering possible mechanisms through which this protein may be degraded by autophagy); and add a discussion of possible implications for different aspects of T cell function, of the regulation of IL-7 by autophagy.

This work, therefore, provides an extremely useful tool to better study autophagy in T (and other) cells and identifies IL-7 receptor as a key substrate of autophagy in T cells that may explain not only how autophagy control T cell proliferation but also other aspects of T cell biology that have been shown to be regulated by autophagy.

This reviewer is satisfied with the new data and discussion.

Reviewer #3 (Remarks to the Author):

I thank the authors for their careful responses to my comments.

This reviewer is satisfied with the new data and discussion.

We are glad that our previous point-to-point reply addressed all three reviewers' concerns. Hope this version of the manuscript will meet your requirements for publication. Thank you very much for your time and consideration. And we are looking forward to your valuable feedback.